# On the Convergence Theory of Debiased Model-Agnostic Meta-Reinforcement Learning

**Alireza Fallah**
EECS Department
Massachusetts Institute of Technology
afallah@mit.edu

**Kristian Georgiev**
EECS Department
Massachusetts Institute of Technology
krisgrg@mit.edu

**Aryan Mokhtari**
ECE Department
The University of Texas at Austin
mokhtari@austin.utexas.edu

**Asuman Ozdaglar**
EECS Department
Massachusetts Institute of Technology
asuman@mit.edu

## Abstract

We consider Model-Agnostic Meta-Learning (MAML) methods for Reinforcement Learning (RL) problems, where the goal is to find a policy using data from several tasks represented by Markov Decision Processes (MDPs) that can be updated by one step of *stochastic* policy gradient for the realized MDP. In particular, using stochastic gradients in MAML update steps is crucial for RL problems since computation of exact gradients requires access to a large number of possible trajectories. For this formulation, we propose a variant of the MAML method, named Stochastic Gradient Meta-Reinforcement Learning (SG-MRL), and study its convergence properties. We derive the iteration and sample complexity of SG-MRL to find an $\epsilon$-first-order stationary point, which, to the best of our knowledge, provides the first convergence guarantee for model-agnostic meta-reinforcement learning algorithms. We further show how our results extend to the case where more than one step of stochastic policy gradient method is used at test time. Finally, we empirically compare SG-MRL and MAML in several deep RL environments.

## 1 Introduction

Meta-learning has recently attracted much attention as a learning to learn approach that enables quick adaptation to new tasks using past experience and data. This is a particularly promising approach for Reinforcement Learning (RL) where in several applications, such as robotics, a group of agents encounter new tasks and need to learn new behaviors or policies through a few interactions with the environment building on previous experience [1–9]. Among various forms of Meta-learning, gradient-based Model-Agnostic Meta-Learning (MAML) formulation [1] is a particularly effective approach which, as its name suggests, can be applied to any learning problem that is trained with gradient-based updates. In MAML, we exploit observed tasks at training time to find an initial model that is trained in a way that rapidly adapts to a new unseen task at test time, after running a few steps of a gradient-based update with respect to the loss of the new task.

The MAML formulation can be extended to RL problems if we represent each task as a Markov Decision Process (MDP). In this setting, we assume that we are given a set of MDPs corresponding to the tasks that we observe during the training phase and assume that the new task at test time is drawn from an underlying probability distribution. The goal in Model-Agnostic Meta-Reinforcement Learning (MAMRL) is to exploit this data to come up with an initial policy that adapts to a new task (drawn from the same distribution) at test time by taking a few stochastic policy gradient steps [1].

35th Conference on Neural Information Processing Systems (NeurIPS 2021).

Several algorithms have been proposed in the context of MAMRL [1, 9–12] which demonstrate the advantage of this framework in practice. None of these methods, however, are supported by theoretical guarantees for their convergence rate or overall sample complexity. Moreover, these methods aim to solve a specific form of MAMRL that does not fully take into account the stochasticity aspect of RL problems. To be more specific, the original MAMRL formulation proposed in [1] assumes performing one step of *policy gradient* to update the initial model at test time. However, as mentioned in the experimental evaluation section in [1], it is more common in practice to use *stochastic* policy gradient, computed over a batch of trajectories, to update the initial model at test time. This is mainly due to the fact that computing the exact gradient of the expected reward is not computationally tractable due to the massive number of possible state-action trajectories. As a result, the algorithm developed in [1] is designed for finding a proper initial policy that performs well after one step of policy gradient, while in practice it is implemented with stochastic policy gradient steps. Due to this difference between the formulation and what is used in practice, the ascent step used in MAML takes a gradient estimate which suffers from a non-diminishing *bias*. As the variance of gradient estimation is also non-diminishing, the resulting algorithm would not achieve exact first-order optimality. To be precise, in stochastic nonconvex optimization, if we use an unbiased gradient estimator, along with a small stepsize or a large batch size to control the variance, the iterates converge to a stationary point. However, if we use a biased estimator with non-vanishing bias and variance, exact convergence to a stationary point is not achievable, even if the variance is small.

**Contributions.** The goal of this paper is to solve the modified formulation of model-agnostic meta-reinforcement learning problem in which we perform a stochastic policy gradient update at test time instead of (deterministic) policy gradient. To do so, we propose a novel stochastic gradient-based method for Meta-Reinforcement Learning (SG-MRL), which is designed for *stochastic policy gradient* steps at test time. We show that SG-MRL implements an *unbiased* estimate of its objective function gradient which allows achieving first-order optimality in non-concave settings. Moreover, we characterize the relation between batch sizes and other problem parameters and the best accuracy that SG-MRL can achieve in terms of gradient norm. We show that, for any $\epsilon > 0$, SG-MRL can find an $\epsilon$-first-order stationary point if the learning rate is sufficiently small or the batch of tasks is large enough. To the best of our knowledge, this is the first result on the convergence of MAMRL methods. Moreover, we show that our analysis can be extended to the case where more than one step of stochastic policy gradient is taken during test time. For simplicity, we state all the results in the body of the paper for the single-step case and include the derivations of the general multiple steps case in the appendices. We also empirically validate the proposed SG-MRL algorithm in larger-scale environments standard in modern reinforcement learning applications, including a 2D-navigation problem, and a more challenging locomotion problem simulated with the MuJoCo library.

**Related work.** Although this paper provides the first theoretical study of MAML for RL, several recent papers have studied the complexity analysis of MAML in other contexts. In particular, the iMAML algorithm which performs an approximation of one step of proximal point method (instead of a few steps of gradient descent) in the inner loop was proposed in [13]. The authors focus on the deterministic case, and show that, assuming the inner loop loss function is sufficiently smooth, i.e., the regularized inner loop function is strongly convex, iMAML converges to a first-order stationary point. Another recent work [14] establishes convergence guarantees of the MAML method to first-order stationarity for non-convex settings. Also, [15] extends the theoretical framework in [14] to the multiple-step case. However, the results in [14, 15] cannot be applied to the reinforcement learning setting. This is mainly due to the fact that *the probability distribution over possible trajectories of states and actions varies with the policy parameter*, leading to a different algorithm that has an additional term which makes the analysis, such as deriving an upper bound on the smoothness parameter, more challenging. We will discuss this point in subsequent sections.

The online meta-learning setting has also been studied in a number of recent works [16–18]. In particular, [17] studies this problem for convex objective functions by casting it in the online convex optimization framework. Also, [16] extends the model-agnostic setup to the online learning case by considering a competitor which adapts to new tasks, and propose the follow the meta leader method which obtains a sublinear regret for strongly convex loss functions.

It is also worth noting that another notion of bias that has been studied in the MAMRL literature [10, 19] differs from what we consider in our paper. More specifically, as we will show later, the derivative of the MAML objective function requires access to the second-order information, i.e., Hessian. In [1], the authors suggest a first-order approximation which ignores this second-order term.

This leads to a biased estimate of the derivative of the MAML objective function, and a number of recent works [10, 19] focus on providing unbiased estimates for the second-order term. In contrast, here we focus on biased gradient estimates where the bias stems from the fact that in most real settings we do not have access to all possible trajectories and we only have access to a mini-batch of possible trajectories. In this case, even if one has access to the second-order term required in the update of MAML, the bias issue we discuss here will remain.

## 2 Problem formulation

Let $\{\mathcal{M}_i\}_i$ be the set of Markov Decision Processes (MDPs) representing different tasks[1]. We assume these MDPs are drawn from a distribution $p$ (which we can only draw samples from), and also the time horizon is fixed and is equal to $\{0, 1, ..., H\}$ for all tasks. For the $i$-th MDP denoted by $\mathcal{M}_i$, which corresponds to task $i$, we denote the set of states and actions by $\mathcal{S}_i$ and $\mathcal{A}_i$, respectively. We also assume the initial distribution over states in $\mathcal{S}_i$ is given by $\mu_i(\cdot)$ and the *transition kernel* is denoted by $P_i$, i.e., the probability of going from state $s \in \mathcal{S}_i$ to $s' \in \mathcal{S}_i$ given taking action $a \in \mathcal{A}_i$ is $P_i(s'|s, a)$. Finally, we assume at state $s$ and by taking action $a$, the agent receives reward $r_i(s, a)$. To summarize, an MDP $\mathcal{M}_i$ is defined by the tuple $(\mathcal{S}_i, \mathcal{A}_i, \mu_i, P_i, r_i)$. For MDP $\mathcal{M}_i$, the actions are chosen according to a *random policy* which is a mixed strategy over the set of actions and depends on the current state, i.e., if the system is in state $s \in \mathcal{S}_i$, the agent chooses action $a \in \mathcal{A}_i$ with probability $\pi_i(a|s)$. To search over the space of all policies, we assume these policies are parametrized with $\theta \in \mathbb{R}^d$, and denote the policy corresponding to parameter $\theta$ by $\pi_i(\cdot|\cdot; \theta)$.

A realization of states and actions in this setting is called a *trajectory*, i.e., a trajectory of MDP $\mathcal{M}_i$ can be written as $\tau = (s_0, a_0, ..., s_H, a_H)$ where $a_h \in \mathcal{A}_i$ and $s_h \in \mathcal{S}_i$ for any $0 \le h \le H$. Note that, given the above assumptions, the probability of this particular trajectory is given by

$$q_i(\tau; \theta) := \mu_i(s_0) \prod_{h=0}^{H} \pi_i(a_h|s_h; \theta) \prod_{h=0}^{H-1} P_i(s_{h+1}|s_h, a_h). \tag{1}$$

Also, the total reward received over this trajectory is $\mathcal{R}_i(\tau) := \sum_{h=0}^{H} \gamma^h r_i(s_h, a_h)$, where $0 \le \gamma \le 1$ is the *discount factor*. As a result, for MDP $\mathcal{M}_i$, the expected reward obtained by choosing policy $\pi(\cdot|\cdot; \theta)$ is given by

$$J_i(\theta) := \mathbb{E}_{\tau \sim q_i(\cdot; \theta)} [\mathcal{R}_i(\tau)]. \tag{2}$$

It is worth noting that the gradient $\nabla J_i(\theta)$ admits the following characterization [20–22]

$$\nabla J_i(\theta) = \mathbb{E}_{\tau \sim q_i(\cdot; \theta)} [g_i(\tau; \theta)], \tag{3}$$

where $g_i(\tau; \theta)$ is defined as

$$g_i(\tau; \theta) := \sum_{h=0}^{H} \nabla_\theta \log \pi_i(a_h|s_h; \theta) \mathcal{R}_i^h(\tau), \tag{4}$$

if we define $\mathcal{R}_i^h(\tau)$ as $\mathcal{R}_i^h(\tau) := \sum_{t=h}^{H} \gamma^t r_i(s_t, a_t)$. In practice, evaluating the exact value of (3) is not computationally tractable. Instead, one could first acquire a batch $\mathcal{D}^{i,\theta}$ of trajectories drawn independently from distribution $q_i(\cdot; \theta)$, and then, estimate $\nabla J_i(\theta)$ by

$$\tilde{\nabla} J_i(\theta, \mathcal{D}^{i,\theta}) := \frac{1}{|\mathcal{D}^{i,\theta}|} \sum_{\tau \in \mathcal{D}^{i,\theta}} g_i(\tau; \theta). \tag{5}$$

Also, we denote the probability of choosing (with replacement) an independent batch of trajectories $\mathcal{D}^{i,\theta}$ by $q_i(\mathcal{D}^{i,\theta}; \theta)$ (see Appendix A.1 for a remark on this).

In this setting, the goal of Model-Agnostic Meta-Reinforcement Learning problem introduced in [1] is to find a good initial policy that performs well in expectation when it is updated using one or a few steps of *stochastic policy gradient* with respect to a new task. In particular, for the case of performing one step of stochastic policy gradient, the problem can be written as[2]

$$\max_{\theta \in \mathbb{R}^d} V_1(\theta) := \mathbb{E}_{i \sim p} \left[ \mathbb{E}_{\mathcal{D}_{test}^i} \left[ J_i \left( \theta + \alpha \tilde{\nabla} J_i(\theta, \mathcal{D}_{test}^i) \right) \right] \right]. \tag{6}$$

---

[1]To simplify the analysis, we assume the number of tasks is finite

[2]From now on, we suppress the $\theta$ dependence of batches to simplify the notation.

Note that by solving this problem we find an initial policy (Meta-policy) that in expectation performs well if we evaluate the output of our procedure after running one step of stochastic policy gradient on this initial policy for a new task.

This formulation can be extended to the setting with more than one step of stochastic policy gradient as well. To state the problem formulation in this case, let us first define $\Psi_i$ which is an operator that takes model $\theta$ and batch $\mathcal{D}^i$ as input and performs one step of stochastic gradient policy at point $\theta$ and with respect to function $J_i$ and batch $\mathcal{D}^i$, i.e., $\Psi_i(\theta, \mathcal{D}^i) := \theta + \alpha \tilde{\nabla} J_i(\theta, \mathcal{D}^i)$. Now, we extend problem (6) to the case where we are looking for an initial point which performs well on expectation after it is updated with $\zeta$ steps of stochastic policy gradient with respect to a new MDP drawn from distribution $p$. This problem can be written as

$$\max_{\theta \in \mathbb{R}^d} V_\zeta(\theta) := \mathbb{E}_{i \sim p}\left[ \mathbb{E}_{\{\mathcal{D}^i_{test,t}\}_{t=1}^\zeta}\left[ J_i\big( \Psi_i(\dots(\Psi_i(\theta, \mathcal{D}^i_{test,1})\dots), \mathcal{D}^i_{test,\zeta}) \big) \right] \right], \tag{7}$$

where the operator $\Psi_i$ is applied $\zeta$ times inside the expectation. In this paper, we establish convergence properties of policy gradient methods for both single step and multiple steps of stochastic gradient cases, but for simplicity in the main text we focus on the single step case.

## 2.1 Second-order information of the expected reward

Due to the inner gradient in $V_1(\theta)$, i.e., the objective function of the MAML problem in (6), the gradient of the function $V_1(\theta)$ requires access to the second-order information of the expected reward function $J(\theta)$. To facilitate further analysis, in this subsection we formally present a characterization of expected reward Hessian and its unbiased estimate over a batch of trajectories. In particular, the expected reward Hessian $\nabla^2 J_i(\theta)$ is given by (see [22] for more details)

$$\nabla^2 J_i(\theta) = \mathbb{E}_{\tau \sim q_i(\cdot;\theta)}\left[ u_i(\tau; \theta) \right], \quad u_i(\tau; \theta) := \nabla_\theta \nu_i(\tau; \theta) \nabla_\theta \log q_i(\tau; \theta)^\top + \nabla_\theta^2 \nu_i(\tau; \theta) \tag{8}$$

where $\nu_i(\tau; \theta)$ is given by $\nu_i(\tau; \theta) := \sum_{h=0}^H \log \pi_i(a_h|s_h; \theta) \mathcal{R}_i^h(\tau)$.

Recall that the reward function is defined as $\mathcal{R}_i^h(\tau) := \sum_{t=h}^H \gamma^t r_i(s_t, a_t)$. It is worth noting that based on the expression in (4) we can write $g_i(\tau; \theta) = \nabla_\theta \nu_i(\tau; \theta)$.

Similar to policy gradient, policy Hessian can be estimated over a batch of trajectories $\mathcal{D}^i$ independently drawn with respect to $q_i(;\theta)$. Specifically, for a given dataset $\mathcal{D}^i$, we can define $\tilde{\nabla}^2 J_i(\theta, \mathcal{D}^i)$

$$\tilde{\nabla}^2 J_i(\theta, \mathcal{D}^i) := \frac{1}{|\mathcal{D}^i|} \sum_{\tau \in \mathcal{D}^i} u_i(\tau; \theta) \tag{9}$$

as an unbiased estimator of the Hessian $\nabla^2 J_i(\theta)$. We will use the expressions for the Hessian $\nabla^2 J_i(\theta)$ in (8) and the Hessian approximation $\tilde{\nabla}^2 J_i(\theta, \mathcal{D}^i)$ in (9) to introduce our proposed method for solving the Meta-RL problem in (6) and its generalized version in (7).

## 3 Model-agnostic meta reinforcement learning

In this section, we first propose a method to solve the stochastic gradient-based MAML Reinforcement Learning problem introduced in (6). Then, we discuss how to extend the proposed method to the setting that we solve a multi-step MAML problem as introduced in (7). We close the section by discussing the differences between our proposed method and the Meta-RL method proposed in [1] and clarify why these two methods are solving two different problems.

### 3.1 MAML for stochastic meta-RL

Our goal in this section is to propose an efficient method for solving the stochastic Meta-RL problem in (6). To do so, we propose a stochastic gradient MAML method for Meta-Reinforcement Learning (SG-MRL) that aims at solving problem (6) by following the update of stochastic gradient descent for the objective function $V_1(\theta)$. To achieve this goal one need to find an unbiased estimator of the gradient $\nabla V_1(\theta)$ which in some MAML settings is not trivial (for more details see Section 4.1 in [14]), but we show that for problem (6) an unbiased estimate of $\nabla V_1(\theta)$ can be efficiently computed.

Let us start by pointing out that the gradient of the function $V_1(\theta)$ defined in (6) is given by

$$\nabla V_1(\theta) = \nabla_\theta \left[ \mathbb{E}_i \, \mathbb{E}_{\mathcal{D}_{test}^i} \left[ J_i \left( \theta + \alpha \tilde{\nabla} J_i(\theta, \mathcal{D}_{test}^i) \right) \right] \right] = \mathbb{E}_i \mathbb{E}_{\mathcal{D}_{test}^i} \left[ (I + \alpha \tilde{\nabla}^2 J_i(\theta, \mathcal{D}_{test}^i)) \right.$$

$$\left. \times \nabla J_i(\theta + \alpha \tilde{\nabla} J_i(\theta, \mathcal{D}_{test}^i)) + J_i(\theta + \alpha \tilde{\nabla} J_i(\theta, \mathcal{D}_{test}^i)) \sum_{\tau \in \mathcal{D}_{test}^i} \nabla_\theta \log \pi_i(\tau; \theta) \right] \tag{10}$$

with the convention that for $\tau = (s_0, a_0, ..., s_H, a_H)$ we define $\pi_i(\tau; \theta)$ as

$$\pi_i(\tau; \theta) := \prod_{h=0}^{H} \pi_i(a_h | s_h; \theta). \tag{11}$$

Recall that the expected reward function $J_i(\theta)$ and its gradient $\nabla J_i(\theta)$ are defined in (2) and (3), respectively, and $\tilde{\nabla} J_i(\theta, \mathcal{D}_{test}^i)$ and $\tilde{\nabla}^2 J_i(\theta, \mathcal{D}_{test}^i)$ are the stochastic estimates of the gradient and Hessian corresponding to $J_i(\theta)$ that are formally defined in (5) and (9), respectively.

Note that the first term in the definition of $\nabla V_1(\theta)$ in (10), i.e., $(I + \alpha \tilde{\nabla}^2 J_i(\theta, \mathcal{D}_{test}^i)) \nabla J_i(\theta + \alpha \tilde{\nabla} J_i(\theta, \mathcal{D}_{test}^i))$, is the term that gives the gradient of an MAML problem (see, e.g., [16]), while the second term, i.e., $J_i(\theta + \alpha \tilde{\nabla} J_i(\theta, \mathcal{D}_{test}^i)) \sum_{\tau \in \mathcal{D}_{test}^i} \nabla_\theta \log \pi_i(\tau; \theta)$, is specific to the RL setting since the probability distribution $p_i$ itself depends on the parameter $\theta$. For more details regarding the derivation $\nabla V_\zeta(\theta)$ for any $\zeta \geq 1$, we refer the reader to Appendix C.

We solve the optimization problem in (6) by using gradient ascent step to update the parameter $\theta$, i.e., following the update $\theta_{k+1} = \theta_k + \beta \nabla V_1(\theta_k)$ at iteration $k$. However, computing the gradient $\nabla V_1(\theta_k)$ may not be tractable in many cases due to the large number of tasks and the size of the action and state spaces. In our proposed SG-MRL method we therefore replace the gradient $\nabla V_1(\theta_k)$ with its estimate computed as follows: At iteration $k + 1$, we first choose a subset $\mathcal{B}_k$ of the tasks (MDPs), where each task is drawn independently from the probability distribution $p$. The SG-MRL outlined in Algorithm 1 is implemented at two levels: (i) inner loop and (ii) outer loop. In the inner loop, for each task $\mathcal{T}_i$ with $i \in \mathcal{B}_k$, we draw a batch of trajectories $\mathcal{D}_{in}^i$ according to $q_i(\cdot; \theta_k)$ to compute the stochastic gradient $\tilde{\nabla} J_i(\theta_k, \mathcal{D}_{in}^i)$ as defined in Section 2. This estimate is then used to compute a model $\theta_{k+1}^i$ corresponding to task $\mathcal{T}_i$ by a single iteration of stochastic policy gradient,

$$\theta_{k+1}^i = \theta_k + \alpha \tilde{\nabla} J_i(\theta_k, \mathcal{D}_{in}^i). \tag{12}$$

For simplicity, we assume that the size of $\mathcal{B}_k$ is equal to $B$ for all $k$, and the size of dataset $\mathcal{D}_{in}^i$ is fixed for all tasks and at each iteration, and we denote it by $D_{in}$.

In the outer loop, we compute the next iterate $\theta_{k+1}$ using the iterates $\{\theta_{k+1}^i\}_{i \in \mathcal{B}_k}$ that are computed in the inner loop. In particular, we follow the update $\theta_{k+1} = \theta_k + \beta \tilde{\nabla} V_1(\theta_k)$, where

$$\tilde{\nabla} V_1(\theta_k) := \frac{1}{B} \sum_{i \in \mathcal{B}_k} \left[ (I + \alpha \tilde{\nabla}^2 J_i(\theta_k, \mathcal{D}_{in}^i)) \tilde{\nabla} J_i(\theta_k + \alpha \tilde{\nabla} J_i(\theta_k, \mathcal{D}_{in}^i), \mathcal{D}_o^i) \right. \tag{13}$$

$$\left. + \tilde{J}_i \left( \theta_k + \alpha \tilde{\nabla} J_i(\theta_k, \mathcal{D}_{in}^i), \mathcal{D}_o^i \right) \sum_{\tau \in \mathcal{D}_{in}^i} \nabla_\theta \log \pi_i(\tau; \theta_k) \right]$$

in which $\tilde{\nabla}^2 J_i(\theta_k, \mathcal{D}_{in}^i)$ is policy Hessian estimate defined in (9) and for each task $\mathcal{T}_i$, the dataset $\mathcal{D}_o^i$ is a new batch of trajectories that are drawn based on the probability distribution $q_i(\cdot; \theta_{k+1}^i)$; Again, for simplicity, we assume that the size of dataset $\mathcal{D}_o^i$ is fixed for all tasks and at each iteration denoted by $D_o$. SG-MRL is summarized in Algorithm 1.

It can be verified that if all the gradients and Hessians in SG-MRL update were exact, then the outcome of the update of SG-MRL would be equivalent to the outcome of gradient ascent update for the function $V_1$, i.e., $\theta_{k+1} = \theta_k + \beta \nabla V_1(\theta_k)$. Note that by computing the expected value of $\tilde{\nabla} V_1(\theta_k)$ first with respect to the random set $\mathcal{D}_o^i$, then with respect to $\mathcal{D}_{in}$, and finally with respect to $\mathcal{B}_k$, we obtain that $\mathbb{E}[\tilde{\nabla} V_1(\theta_k)] = \nabla V_1(\theta_k)$. Therefore, the stochastic gradient $\tilde{\nabla} V_1(\theta_k)$ is an unbiased estimator of the gradient $\nabla V_1(\theta_k)$.

The SG-MRL method can also be extended and used for solving the multi-step MAML problem defined in (7). To do so, at each iteration, we first perform $\zeta$ steps of policy stochastic gradient in the

**Algorithm 1:** Proposed SG-MRL method for Meta-RL

---

**Input:** Initial iterate $\theta_0$

**repeat**

    Draw a batch of i.i.d. tasks $\mathcal{B}_k \subseteq \mathcal{I}$ with size $B = |\mathcal{B}_k|$;

    **for** all $\mathcal{T}_i$ with $i \in \mathcal{B}_k$ **do**

        Sample a batch of trajectories $\mathcal{D}_{in}^i$ w.r.t. $q_i(\cdot; \theta_k)$;

        Set $\theta_{k+1}^i = \theta_k + \alpha \tilde{\nabla} J_i(\theta_k, \mathcal{D}_{in}^i)$;

    **end for**

    Sample a batch of trajectories $\mathcal{D}_o^i$ w.r.t. $q_i(\cdot; \theta_{k+1}^i)$;

    Set $\theta_{k+1} = \theta_k$

$$+ \frac{\beta}{B} \sum_{i \in \mathcal{B}_k} \left( \left( I + \alpha \tilde{\nabla}^2 J_i(\theta_k, \mathcal{D}_{in}^i) \right) \tilde{\nabla} J_i\left( \theta_{k+1}^i, \mathcal{D}_o^i \right) + \overbrace{\tilde{J}_i\left( \theta_{k+1}^i, \mathcal{D}_o^i \right) \sum_{\tau \in \mathcal{D}_{in}^i} \nabla_\theta \log \pi_i(\tau; \theta_k)}^{\text{Additional term in SG-MRL}} \right)$$

    $k \leftarrow k + 1$

**until** not done

---

inner loop, and then take one step of stochastic gradient ascent with respect to an unbiased estimator of $\nabla V_\zeta(\theta)$. More details on the implementation of SG-MRL for that case is provided in Appendix C.

## 3.2 Comparing SG-MRL with other model-agnostic meta-RL methods

In this section, we discuss the difference between our SG-MRL method and recent Meta-RL methods. In particular, we focus on the MAML method in [1] for solving RL problems. Before discussing the differences between these two methods, let us first recap the update of the MAML method in [1].

The main formulation proposed in [1] which was followed in other works such as [10] is slightly different from the one in this paper as they assume the agent has access to the *exact gradient* of the new task, and hence, they consider the following MAML problem

$$\max_{\theta \in \mathbb{R}^d} \hat{V}_1(\theta) := \mathbb{E}_{i \sim p} \left[ J_i \left( \theta + \alpha \nabla J_i(\theta) \right) \right]. \tag{14}$$

As mentioned, the main difference between (6) and (14) is that the former tries to find a good initial policy that leads to a good solution after running one step of *stochastic gradient ascent*, while the latter finds an initial policy that produces a good policy after running one step of *gradient ascent*.

**Remark 1.** *Problems in (6) and (14) are both valid formulations for Meta-RL. In practice, however, it is often computationally intractable to evaluate the exact gradient of the expected reward and we often have only access to its stochastic gradient. Hence, it might be more practical to solve (6) instead of (14) as it finds an initial policy that performs well after running one step of stochastic gradient, unlike (14) that finds a policy that performs well after running one step of gradient update.*

In a nutshell, the MAML method proposed in [1] tries to solve the problem in (14) by following the update of stochastic gradient ascent for the objective function $\hat{V}_1(\theta)$. To be more precise, note that the gradient of the loss function $\hat{V}_1(\theta)$ defined in (14) can be expressed as

$$\nabla \hat{V}_1(\theta) = \nabla_\theta \mathbb{E}_{i \sim p} \left[ J_i \left( \theta + \alpha \nabla J_i(\theta) \right) \right] = \mathbb{E}_{i \sim p} \left[ \left( I + \alpha \nabla^2 J_i(\theta) \right) \nabla J_i \left( \theta + \alpha \nabla J_i(\theta) \right) \right]. \tag{15}$$

Note that the expression for the gradient of $\hat{V}_1(\theta)$ in (15) is different from the expression for the gradient of $V_1(\theta)$ in (10). In particular, the extra term $J_i(\theta + \alpha \tilde{\nabla} J_i(\theta, \mathcal{D}_{test}^i)) \sum_{\tau \in \mathcal{D}_{test}^i} \nabla_\theta \log \pi_i(\tau; \theta)$ that appears in (15) is caused by the fact that we use stochastic gradients in the definition of the function $V_1(\theta)$, while exact gradients are used in the definition of $\hat{V}_1(\theta)$.

Considering the expression for the gradient of $\hat{V}_1(\theta)$ in (15), a natural approach to approximate $\nabla \hat{V}_1(\theta)$ is to replace the gradients and Hessians corresponding to the expected reward $J_i(\theta)$ by their stochastic approximations. In other words, one can use the approximation $\tilde{\nabla} \hat{V}_1(\theta_k)$ which is defined as the average over $(I + \alpha \tilde{\nabla}^2 J_i(\theta_k, \mathcal{D}_{in}^i)) \tilde{\nabla} J_i(\theta_k + \alpha \tilde{\nabla} J_i(\theta_k, \mathcal{D}_{in}^i), \mathcal{D}_o^i)$ for all $i \in \mathcal{B}_k$, i.e.,

$$\tilde{\nabla} \hat{V}_1(\theta_k) := \frac{1}{B} \sum_{i \in \mathcal{B}_k} \left( I + \alpha \tilde{\nabla}^2 J_i(\theta_k, \mathcal{D}_{in}^i) \right) \tilde{\nabla} J_i \left( \theta_k^i, \mathcal{D}_o^i \right) \tag{16}$$

where $\theta_k^i := \theta_k + \alpha \tilde{\nabla} J_i(\theta_k, \mathcal{D}_{in}^i)$. Here the procedure for computing the sample sets $\mathcal{D}_{in}^i$ and $\mathcal{D}_o^i$ is the same as the one in SG-MRL. Once $\tilde{\nabla} \hat{V}_1(\theta_k)$ is computed the new variable $\theta_{k+1}$ can be computed by following the update of stochastic gradient ascent, i.e., $\theta_{k+1} = \theta_k + \beta \ \tilde{\nabla} \hat{V}_1(\theta_k)$. The description of the Meta-RL method in [1] and its implementation at two levels (inner and outer) is similar to the one in Algorithm 1, except the highlighted additional term which is not included in MAML update.

Note that the gradient estimate $\tilde{\nabla} \hat{V}_1(\theta_k)$ in (16) is a *biased* estimate of the exact gradient $\nabla \hat{V}_1(\theta_k)$ defined in (15). This is due to the fact that $\tilde{\nabla} J_i(\theta_k + \alpha \tilde{\nabla} J_i(\theta_k, \mathcal{D}_{in}^i), \mathcal{D}_o^i)$ is a biased estimate of $\nabla J_i(\theta + \alpha \nabla J_i(\theta))$ because of the term $\tilde{\nabla} J_i(\theta_k, \mathcal{D}_{in}^i)$ inside it. In other words, MAML method proposed by [1] uses a biased estimate of the gradient in this case. Note that, in general optimization analyses, when we have access to biased gradient estimators, even with diminishing or small stepsize, we might only converge to a neighborhood of the optimal solution, where the radius of our convergence depends on the bias. To resolve this issue, one needs to control the bias in the gradient directions and lower the bias as time progresses using some debiasing techniques. For instance, the work in [23] studies this problem in detail for debiasing MAML in the supervised learning setting.

On the other hand, our proposed SG-MRL method does not suffer from this issue since computing an unbiased estimator of the gradient for the objective function considered in (6) is relatively simple. In fact, in the following section, we show that SG-MRL is provably convergent and characterize its complexity to find an approximate first-order stationary point of (6) and its generalized version defined in (7).

## 4 Theoretical results

In this section, we study the convergence properties of the proposed SG-MRL method and characterize its overall complexity for finding a policy that satisfies the first-order optimality condition for the objective function $V_\zeta(\theta)$ defined in (7). To do so, we first formally define the first-order optimality condition that we aim to achieve.

**Definition 1.** *A random vector $\theta_\epsilon \in \mathbb{R}^d$ is called an $\epsilon$-approximate first-order stationary point (FOSP) for problem* (7) *if it satisfies* $\mathbb{E}[\|\nabla V_\zeta(\theta_\epsilon)\|] \leq \epsilon$.

We next state the main assumptions that we use to derive our results.

**Assumption 1.** *The reward functions $r_i$ are nonnegative and uniformly bounded, i.e., there exists a constant $R$ such that for any task $i$, state $s \in \mathcal{S}_i$, and action $a \in \mathcal{A}_i$, we have $0 \leq r_i(a|s) \leq R$.*

**Assumption 2.** *There exist constants $G$ and $L$ such that for any $i$ and for any state $s \in \mathcal{S}_i$, action $a \in \mathcal{A}_i$, and parameter $\theta \in \mathbb{R}^d$, we have $\|\nabla_\theta \log \pi_i(a|s; \theta)\| \leq G$ and $\|\nabla_\theta^2 \log \pi_i(a|s; \theta)\| \leq L$.*

Both assumptions are customary in the policy gradient literature and have been used in other papers to obtain convergence guarantees for policy gradient methods [24, 22, 25].

**Assumption 3.** *There exists a constant $\rho$ such that for any $i$ and for any state $s \in \mathcal{S}_i$, action $a \in \mathcal{A}_i$, and parameters $\theta_1, \theta_2 \in \mathbb{R}^d$, we have $\|\nabla_\theta^2 \log \pi_i(a|s; \theta_1) - \nabla_\theta^2 \log \pi_i(a|s; \theta_2)\| \leq \rho \|\theta_1 - \theta_2\|$.*

This assumption is also customary in the analysis of MAML-type algorithms [14, 16]. In particular, in Appendix B we provide more insight into the conditions in Assumptions 2 and 3 by focusing on the special case of *softmax policy parametrization*.

### 4.1 Convergence of SG-MRL

Next, we study the convergence of our proposed SG-MRL for solving the Model-Agnostic Meta-Reinforcement Learning problem in (7). To do so, we show two important intermediate results. First, we show that the function $V_\zeta(\theta)$ is smooth. Second, we show the unbiased estimator of the gradient $\nabla V_\zeta(\theta)$ denoted by $\tilde{\nabla} V_\zeta(\theta_k)$ has a bounded norm. Building on these two results, we will derive the convergence of SG-MRL. To prove these two intermediate results, we first state the following lemma on the Lipschitz property of the expected reward function $J_i$ and its first and second derivatives for any MDP $\mathcal{M}_i$. This lemma not only plays a key role in our analysis, but also can be of independent interest in general for analyzing meta-reinforcement learning algorithms.

**Lemma 1.** *Recall the definitions of $g_i(\tau; \theta)$ in (4) and $u_i(\tau; \theta)$ in (8) for trajectory $\tau \in (\mathcal{S}_i \times \mathcal{A}_i)^{H+1}$ and policy parameter $\theta \in \mathbb{R}^d$. If Assumptions 1-3 hold, then for any MDP $\mathcal{M}_i$ we have:*

*i) For any $\tau$ and $\theta$, we have $\|g_i(\tau;\theta)\| \leq \eta_G := \frac{GR}{(1-\gamma)^2}$. As a consequence, $\|\nabla J_i(\theta)\|, \|\tilde{\nabla} J_i(\theta, \mathcal{D}^i)\| \leq \eta_G$ for any $\theta$ and any batch of trajectories $\mathcal{D}^i$. Further, this implies that $J_i(.)$ is smooth with parameter $\eta_G$.*

*ii) For any $\tau$ and $\theta$, we have $\|u_i(\tau;\theta)\| \leq \eta_H := \frac{((H+1)G^2+L)R}{(1-\gamma)^2}$. As a consequence, $\|\nabla^2 J_i(\theta)\|, \|\tilde{\nabla}^2 J_i(\theta, \mathcal{D}^i)\| \leq \eta_H$ for any $\theta$ and any batch of trajectories $\mathcal{D}^i$. Further, this implies that $\nabla J_i(.)$ is smooth with parameter $\eta_H$.*

*iii) For any batch of trajectories $\mathcal{D}^i$, $\tilde{\nabla}^2 J_i(\theta, \mathcal{D}^i)$ is smooth with parameter $\eta_\rho := \frac{(2(H+1)GL+\rho)R}{(1-\gamma)^2}$.*

By exploiting the results in Lemma 1, we can prove the promised results on the Lipschitz property of $\nabla V_\zeta(\theta)$ as well as boundedness of its unbiased estimator $\tilde{\nabla} V_\zeta(\theta)$. In the following proposition, due to space limitation and for the the ease of notation we only state the result for the case that $\zeta = 1$; however, the general version of these results along with their proofs are available in Appendix F.

**Proposition 1.** *Consider the objective function $V_1$ defined in (6) for the case that $\alpha \in (0, 1/\eta_H]$ where $\eta_H$ is given in Lemma 1. Suppose that the conditions in Assumptions 1-3 are satisfied. Then,*

*i) $V_1(\theta)$ is smooth with parameter*

$$L_V := \alpha\eta_\rho\eta_G + 4\eta_H + 8RD_{in}(H+1)(L + D_{in}G^2(H+1)) \tag{17}$$

*where $\eta_G$ and $\eta_\rho$ are defined in Lemma 1.*

*ii) For any choices of $\mathcal{B}_k$, $\{\mathcal{D}_o^i\}_i$ and $\{\mathcal{D}_{in}^i\}_i$, the norm of stochastic gradient $\tilde{\nabla} V_1(\theta_k)$ defined in (13) at iteration $k$ is bounded above by $\|\tilde{\nabla} V_1(\theta_k)\| \leq G_V := 2GR[(1-\gamma)^{-2} + D_{in}(H+1)]$.*

The smoothness parameter for the RL problem has been previously characterized (as an example see [22]), but, to the best of our knowledge, this is the first result on the smoothness parameter of the meta-RL function. Proving Proposition 1 is the main challenge in our analysis, since it establishes that our formulation satisfies the relevant assumptions needed for our main result in the next theorem.

Now, we present our main result on the convergence of SG-MRL to a first-order stationary point for the Meta-reinforcement learning problem in defined (7). We state our main result for the special case of $\zeta = 1$, but the general statement of the theorem along with its proof can be found in Appendix G.

**Theorem 1.** *Consider $V_1$ defined in (6) for the case that $\alpha \in (0, 1/\eta_H]$ where $\eta_H$ is defined in Lemma 1. Suppose Assumptions 1-3 are satisfied, and recall the definitions of $L_V$ and $G_V$ from Proposition 1. Consider running SG-MRL (Algorithm 1) with $\beta \in (0, 1/L_V]$. Then, for any $1 > \epsilon > 0$, SG-MRL finds a solution $\theta_\epsilon$ such that $\mathbb{E}[\|\nabla V_1(\theta_\epsilon)\|^2] \leq \frac{2G_V^2 L_V \beta}{BD_o} + \epsilon^2$, after running for at most $\mathcal{O}(1)\frac{R}{\beta} \min\left\{\frac{1}{\epsilon^2}, \frac{BD_o}{G_V^2 L_V \beta}\right\}$ iterations.*

Next we characterize the complexity of SG-MRL for finding an $\epsilon$-first-order stationary point solution.

**Corollary 1.** *Suppose the hypotheses of Theorem 1 hold. Then, for any $\epsilon > 0$, SG-MRL achieves $\epsilon$-first-order stationarity by setting: (i) $BD_o \geq 8G_V^2/\epsilon^2$ and $\beta = 1/L_V$ requiring $\mathcal{O}(\epsilon^{-2})$ iterations and computing $\mathcal{O}(\epsilon^{-2})$ stochastic gradients per iteration; or (ii) $\beta = \mathcal{O}(\epsilon^{-2})$ and $BD_o = \mathcal{O}(1)$ which requires $\mathcal{O}(\epsilon^{-4})$ iterations and $\mathcal{O}(1)$ stochastic gradient evaluations per iteration.*

The conditions in Corollary 1 identify two settings under which SG-MRL finds an $\epsilon-$FOSP after a finite number of iterations, abd both settings overall require $\mathcal{O}(\epsilon^{-4})$ stochastic gradient evaluations.

**Remark 2.** *While we mainly focused on the case $\zeta = 1$, we provide the general statement of the results for any $\zeta$ in the Appendix. Note that the downside of increasing $\zeta$ is that the smoothness parameter grows exponentially with respect to $\zeta$ (see Theorem 3), which means that we need to take a smaller learning rate that leads to a slower convergence rate. However, on the positive side, by increasing $\zeta$ we train a model that better adapts to a new task.*

## 5  Numerical experiments

In this section, we empirically validate the proposed SG-MRL algorithm in larger-scale environments standard in modern reinforcement learning applications. The code is available online[3].

---

[3]The code is available at https://github.com/kristian-georgiev/SGMRL.

Table 1: Mean meta-test reward (negative square distance to goal location) of SG-MRL, MAML, and E-MAML after 1 adaptation step.

| Algorithm | Meta-Test Reward |
|-----------|------------------|
| **SG-MRL** | $\mathbf{-16.901 \pm 0.699}$ |
| MAML | $-17.767 \pm 0.106$ |
| E-MAML | $-17.803 \pm 0.115$ |

Table 2: The mean meta-test reward for SG-MRL and MAML on additional environments when trained and adapted with 1, 2, and 3 inner updates over 4 random seeds.

| environment | SG-MRL reward | MAML reward |
|-------------|---------------|-------------|
| Half-Cheetah Random Direction, 1 step | $\mathbf{580.143 \pm 38.22}$ | $465.624 \pm 54.07$ |
| Half-Cheetah Random Direction, 2 step | $\mathbf{580.203 \pm 33.63}$ | $441.247 \pm 58.34$ |
| Half-Cheetah Random Direction, 3 step | $\mathbf{504.747 \pm 45.07}$ | $477.086 \pm 64.71$ |
| Half-Cheetah Random Velocity, 1 step | $\mathbf{-91.73 \pm 0.34}$ | $-92.92 \pm 0.70$ |
| Half-Cheetah Random Velocity, 2 step | $\mathbf{-52.64 \pm 6.86}$ | $-56.71 \pm 6.73$ |
| Half-Cheetah Random Velocity, 3 step | $-33.39 \pm 0.67$ | $\mathbf{-32.48 \pm 0.50}$ |
| Swimmer Random Velocity, 1 step | $\mathbf{118.77 \pm 9.99}$ | $104.53 \pm 24.18$ |
| Swimmer Random Velocity, 2 step | $\mathbf{134.57 \pm 1.67}$ | $108.47 \pm 23.36$ |
| Swimmer Random Velocity, 3 step | $\mathbf{110.91 \pm 12.56}$ | $90.60 \pm 14.99$ |

We conduct two experiments: a 2D-navigation problem, and a more challenging locomotion problem simulated with the MuJoCo library [26]. For both experiments, we use a neural network policy with a standard feed-forward neural network and optimize it with vanilla policy gradient [27]. Further implementation details are outlined in Appendix H.

All experiments were conducted in MIT's Supercloud [28]. Similar to FO-MAML proposed in [1], we use first order implementation of SG-MRL. It is also worth noting that SG-MRL is straightforward to implement as a modification to MAML and requires no additional hyperparameter tuning. Also, SG-MRL does not reduce the scalability of MAML. In particular, across experiments, we benchmarked the clock time of SG-MRL against MAML and SG-MRL is consistently at most 1.05 times slower over the course of training. Next, we demonstrate the practicality of SG-MRL in modern deep reinforcement learning problems.

**2D-navigation.** We consider the problem of a point-mass agent navigating from the origin to a random goal location within a unit-size square centered at the origin ($[-0.5, 0.5] \times [-0.5, 0.5]$). We consider the negative squared distance to the goal location as a reward. Observations consist of the position of the agent within the unit-size square. The action space comprises of all velocities with components clipped in the interval $[-0.1, 0.1]$. An example of a trajectory is illustrated in Figure 1. In Table 1, we compare the performance of SG-MRL against MAML [1] and E-MAML [29]. We make a comparison with E-MAML since it has a similar spirit to our proposed SG-MRL method, but unlike the proposed algorithm, E-MAML is derived from heuristic arguments.

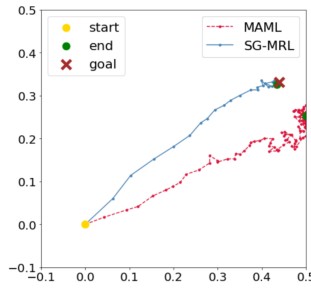

Figure 1: Trajectories generated by policies trained with SG-MRL and MAML for the 2D-navigation problem.

**Locomotion: MuJoCo environments.** In addition to the 2D-navigation example, we provide a benchmark on a more challenging set of tasks - MuJoCo's locomotion environments. We benchmark our algorithm against MAML on three different tasks and report the results in Table 2. The tasks involve learning to move in a goal direction (forward/backward), or reach a target velocity. We describe each task in more detail in Appendix H.

## 6 Conclusion and future work

We studied MAML for RL problems, considering performing a few steps of stochastic policy gradient at test time. Given this formulation, we introduced SG-MRL, and discussed how it differs from the

original MAML algorithm in [1]. Further, we characterized the convergence of SG-MRL method in terms of gradient norm and under a set of assumptions on the policy and reward functions. Our results show that, for any $\epsilon$, SG-MRL achieves $\epsilon$-first-order stationarity, given that either the learning rate is small enough or the multiplication of task and outer loop batch sizes is sufficiently large.

A shortcoming of our analysis is the requirement on the boundedness of gradient norm (Assumption 2). A natural extension of our work would be extending the theoretical results to the setting that gradient norm is possibly unbounded. Moreover, our results are limited to achieving first-order optimality, while one can exploit techniques for escaping from saddle points to obtain second-order stationarity.

## 7  Acknowledgment

Alireza Fallah acknowledges support from the Apple Scholars in AI/ML PhD fellowship and the MathWorks Engineering Fellowship. This research is sponsored by the United States Air Force Research Laboratory and the United States Air Force Artificial Intelligence Accelerator and was accomplished under Cooperative Agreement Number FA8750-19-2-1000. The views and conclusions contained in this document are those of the authors and should not be interpreted as representing the official policies, either expressed or implied, of the United States Air Force or the U.S. Government. The U.S. Government is authorized to reproduce and distribute reprints for Government purposes notwithstanding any copyright notation herein. This research of Aryan Mokhtari is supported in part by NSF Grant 2007668, ARO Grant W911NF2110226, the Machine Learning Laboratory at UT Austin, and the NSF AI Institute for Foundations of Machine Learning.

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
