# A  Intermediate Results

## A.1  A Remark on the Batch of Trajectories

Recall that $q_i(\mathcal{D}^i; \theta)$ denotes the probability of independently drawing batch $\mathcal{D}^i$ of trajectories with respect to $i$-th MDP and at policy parameter $\theta$. Also, as we stated in Section 2, we assume the batch of trajectories are sampled with replacement. Note that, in this case

$$q_i(\mathcal{D}^i; \theta) = \prod_{\tau \in \mathcal{D}^{i,\theta}} q_i(\tau; \theta). \tag{18}$$

However, for the case that the batch of trajectories that we draw is not ordered, we have

$$q_i(\mathcal{D}^i; \theta) = C_{\mathcal{D}^i} \prod_{\tau \in \mathcal{D}^{i,\theta}} q_i(\tau; \theta). \tag{19}$$

with

$$C_{\mathcal{D}^i} = |\mathcal{D}^i|! / \prod_{\tau \in (\mathcal{S}_i \times \mathcal{A}_i)^{H+1}} C_\tau!$$

where $C_\tau$ is the number of times that the particular trajectory $\tau$ is appeared in $\mathcal{D}^i$. Throughout the proofs, we mainly refer to (18). However, the results can be easily extended to (19) as well. The reason is that we mostly work with the term $\nabla_\theta \log q_i(\mathcal{D}^i; \theta)$, and since $C_{\mathcal{D}^i}$ is not a function of $\theta$, for both cases we have

$$\nabla_\theta \log q_i(\mathcal{D}^i; \theta) = \sum_{\tau \in \mathcal{D}^i} \nabla_\theta \log q_i(\tau; \theta) = \sum_{\tau \in \mathcal{D}^i} \nabla_\theta \log \pi_i(\tau; \theta)$$

where the last equality is obtained using (1) along with the definition (11).

## A.2  Lemmas

**Lemma 2.** *For any $i \in \{1, ..., n\}$, let $f_i : \mathbb{R}^d \to W_i$ be a continuous function with $W_i \in \{\mathbb{R}, \mathbb{R}^d, \mathbb{R}^{1 \times d}, \mathbb{R}^{d \times d}\}$ such that $g(\theta) = f_n(\theta)...f_1(\theta)$ is well defined. Furthermore, assume that for any $i$, the following holds:*

1. *$f_i$ is bounded, i.e., $\|f_i(\theta)\| \leq B_i$ for some nonnegative constant $B_i$ and any $\theta \in \mathbb{R}^d$.*

2. *$f_i$ is Lipschitz, i.e., $\|f_i(\theta) - f_i(\tilde{\theta})\| \leq L_i \|\theta - \tilde{\theta}\|$ for some nonnegative constant $L_i$ and any $\theta, \tilde{\theta} \in \mathbb{R}^d$.*

*Then, $g(\theta)$ is Lipschitz with parameter $L_g := \sum_{i=1}^n (L_i \prod_{j \neq i} B_j)$, i.e., for any $\theta$ and $\tilde{\theta}$,*

$$\|g(\theta) - g(\tilde{\theta})\| \leq L_g \|\theta - \tilde{\theta}\|. \tag{20}$$

*Proof.* We prove this result by induction on $n$. First, for $n = 2$, note that

$$
\begin{aligned}
\|g(\theta) - g(\tilde{\theta})\| &= \left\| f_2(\theta) f_1(\theta) - f_2(\tilde{\theta}) f_1(\tilde{\theta}) \right\| \\
&= \left\| f_2(\theta) f_1(\theta) - f_2(\theta) f_1(\tilde{\theta}) + f_2(\theta) f_1(\tilde{\theta}) - f_2(\tilde{\theta}) f_1(\tilde{\theta}) \right\| \\
&\leq \left\| f_2(\theta) f_1(\theta) - f_2(\theta) f_1(\tilde{\theta}) \right\| + \left\| f_2(\theta) f_1(\tilde{\theta}) - f_2(\tilde{\theta}) f_1(\tilde{\theta}) \right\| \\
&\leq \|f_2(\theta)\| \|f_1(\theta) - f_1(\tilde{\theta})\| + \|f_1(\tilde{\theta})\| \|f_2(\theta) - f_2(\tilde{\theta})\| \\
&\leq B_2 L_1 \|\theta - \tilde{\theta}\| + B_1 L_2 \|\theta - \tilde{\theta}\| = L_g \|\theta - \tilde{\theta}\| \tag{21}
\end{aligned}
$$

where the last inequality follows from the boundedness and Lipschitz property assumptions on $f_i$. Next, for $n \geq 3$, we assume the results holds for $n - 1$, and we show it also holds for $n$. Note that if $f_n(\theta)...f_1(\theta)$ is well defined, $f_m(\theta)...f_1(\theta)$ is also well defined for any $m \leq n$, including $m = n - 1$. Hence, by induction hypothesis

$$\|f_{n-1}(\theta)...f_1(\theta) - f_{n-1}(\tilde{\theta})...f_1(\tilde{\theta})\| \leq \tilde{L}_g \|\theta - \tilde{\theta}\|. \tag{22}$$

where $\tilde{L}_g = \sum_{i=1}^{n-1}(L_i \prod_{j \neq i} B_j)$. Thus, $\tilde{g}(\theta) := f_{n-1}(\theta)...f_1(\theta)$ is Lipschitz with parameter $\tilde{L}_g$. Also, it is bounded by $\prod_{j=1}^{n-1} B_j$. Finally, note that $\tilde{g}$ is a function from $\mathbb{R}^d$ to one of $\{\mathbb{R}, \mathbb{R}^d, \mathbb{R}^{1 \times d}, \mathbb{R}^{d \times d}\}$. Thus, using (21), we obtain

$$\|g(\theta) - g(\tilde{\theta})\| = \left\| f_n(\theta)\tilde{g}(\theta) - f_n(\tilde{\theta})\tilde{g}(\tilde{\theta}) \right\| \leq (B_n \tilde{L}_g + L_n \prod_{j=1}^{n-1} B_j)\|\theta - \tilde{\theta}\|. \tag{23}$$

However, it is easy to verify that in fact $B_n \tilde{L}_g + L_n \prod_{j=1}^{n-1} B_j = L_g$ and hence the proof is complete. $\qquad\square$

**Lemma 3.** *For any $i \in \{1, ..., n\}$, let $f_i : \mathbb{R}^d \to \mathbb{R}^m$ be a continuously differentiable function which is bounded by $B_f$, and is also Lipschitz with Lipschitz parameter $L_f$. Also, let $p(.; \theta)$ be a distribution on $\{f_i\}_{i=1}^n$ where probability of drawing $f_i$ is $p(i; \theta)$. We further assume there exists a non-negative constant $B_p$ such that for any $i$ and $\theta$*

$$\|\nabla_\theta \log p(i; \theta)\| \leq B_p. \tag{24}$$

*Then, the function $g(\theta) := \mathbb{E}_{p(i;\theta)}[f(i; \theta)]$ is Lipschitz with parameter $B_f B_p + L_f$.*

*Proof.* First note that

$$\|\nabla_\theta p(i; \theta)\| = \|\nabla_\theta \log p(i; \theta)\| p(i; \theta) \leq B_p \, p(i; \theta). \tag{25}$$

To show the result, it suffices to prove

$$\|\frac{\partial}{\partial \theta} g(\theta)\| \leq B_f B_p + L_f. \tag{26}$$

To show this, note that, by product rule, we have

$$\frac{\partial}{\partial \theta} g(\theta) = \frac{\partial}{\partial \theta}(\sum_i f(i; \theta)p(i; \theta)) = \sum_i p(i; \theta)\frac{\partial}{\partial \theta}f(i; \theta) + \sum_i \nabla p(i; \theta)f(i; \theta)^\top. \tag{27}$$

As a result

$$\begin{aligned}
\|\frac{\partial}{\partial \theta} g(\theta)\| &\leq \sum_i p(i; \theta)\|\frac{\partial}{\partial \theta}f(i; \theta)\| + \sum_i \|\nabla p(i; \theta)\|\|f(i; \theta)\| \\
&\leq L_f \sum_i p(i; \theta) + B_f B_p \sum_i p(i; \theta) \\
&= L_f + B_f B_p
\end{aligned} \tag{28}$$

where first part of (28) follows from the fact that $\|\frac{\partial}{\partial \theta}f(i; \theta)\| \leq L_f$ as $f(i; \theta)$ is Lipschitz with parameter $L_f$, and the second part of (28) is obtained using (25) along with boundedness assumption of $f_i$ functions. $\qquad\square$

## B  Softmax Policy

Consider the function $\phi : \mathcal{A} \times \mathcal{S} \to \mathbb{R}^d$ as an arbitrary mapping from the space of actions-states to real-valued vectors with dimension $d$ which is the size of policy parameter $\theta$. Then, the softmax policy is given by[4]

$$\pi(a|s, \theta) = \frac{\exp(\phi(a, s)^\top \theta)}{\sum_{a' \in \mathcal{A}} \exp(\phi(a', s)^\top \theta)}.$$

In this case, $\nabla_\theta \log \pi(a|s; \theta)$, which is known as the score function, admits the following characterization (see [20])

$$\nabla_\theta \log \pi(a|s; \theta) = \phi(a, s) - \mathbb{E}_{a' \sim \pi(a'|s, \theta)}[\phi(a', s)]. \tag{29}$$

---

[4]Through this example we suppress the task indices and mostly focus on softmax parametrization.

Using this expression, we can show that the Hessian $\nabla^2_\theta \log \pi(a|s;\theta)$ is equal to the negative of covariance matrix of random variable $\phi(a', s)$ when $a'$ is drawn from distribution $\pi(a'|s,\theta)$, i.e.,

$$\nabla^2_\theta \log \pi(a|s;\theta)$$
$$= -\mathbb{E}_{a'\sim\pi(a'|s,\theta)}\left[\left(\phi(a', s) - \mathbb{E}_{a''\sim\pi(a''|s,\theta)}[\phi(a'', s)]\right)\right.$$
$$\left.\left(\phi(a', s) - \mathbb{E}_{a''\sim\pi(a''|s,\theta)}[\phi(a'', s)]\right)^\top\right].$$

For more details regarding the derivation of $\nabla^2_\theta \log \pi(a|s;\theta)$ please check Appendix D.

According to the expressions for $\nabla_\theta \log \pi(a|s;\theta)$ and $\nabla^2_\theta \log \pi(a|s;\theta)$, when we use a softmax policy, if we assume that the mapping norm $\|\phi(.,.)\|$ is bounded, then both conditions in Assumption 2 hold, i.e., $\|\nabla_\theta \log \pi(a|s;\theta)\|$ and $\|\nabla^2_\theta \log \pi(a|s;\theta)\|$ would be both bounded for any action $a$, state $s$, and parameter $\theta$. Moreover, in Appendix D, we further show that the boundedness of $\|\phi(.,.)\|$ implies that the condition in Assumption 3 holds as well.

Hence, at least for the softmax policy, the conditions in Assumptions 2 and 3 hold, if the mapping $\phi$ has a bounded norm. Note that in most applications, the mapping $\phi$ is a neural network and as the weights of neural networks are often bounded (or enforced to be bounded), $\|\phi(.,.)\|$ is uniformly upper bounded.

## C  Multi-Step SG-MRL Method

We first start by characterizing $\nabla V_\zeta(\theta)$ for general $\zeta \geq 1$.

**Theorem 2.** *Recall the definition of $V_\zeta(\theta)$ (7). Then, its derivative can be expressed as*

$$\nabla V_\zeta(\theta) = \mathbb{E}_{i\sim p}\mathbb{E}_{\{\mathcal{D}^i_{test,j}\}^\zeta_{t=1}}\left[\prod^\zeta_{t=1}(I + \alpha\tilde{\nabla}^2 J_i(\theta^{i,t-1}(\theta), \mathcal{D}^i_{test,t'}))\nabla J_i(\theta^{i,\zeta}(\theta))\right.$$
$$\left. + J_i\left(\theta^{i,\zeta}(\theta)\right)\sum^\zeta_{t=1}\left(\prod^{t-1}_{t'=1}(I + \alpha\tilde{\nabla}^2 J_i(\theta^{i,t'-1}(\theta), \mathcal{D}^i_{test,t'}))\sum_{\tau\in\mathcal{D}^i_{test,t}}\nabla_\theta \log \pi_i(\tau; \theta^{i,t-1}(\theta))\right)\right].$$
$$(30)$$

*Proof.* To simplify the notation, let us define $\theta^{i,0}(\theta) := \theta$ and $\theta^{i,t}(\theta) := \Psi_i(...(\Psi_i(\theta, \mathcal{D}^i_{test,1})...), \mathcal{D}^i_{test,t})$ for $t \geq 1$. Then, $V_\zeta(\theta)$ can be cast as

$$V_\zeta(\theta) = \mathbb{E}_{i\sim p}\left[\mathbb{E}_{\{\mathcal{D}^i_{test,t}\}^\zeta_{t=1}}\left[J_i\left(\theta^{i,\zeta}(\theta)\right)\right]\right]. \tag{31}$$

Note that

$$\frac{\partial}{\partial\theta}\Psi_i(\theta, \mathcal{D}^i) = I + \alpha\tilde{\nabla}^2 J_i(\theta, \mathcal{D}^i). \tag{32}$$

Now, using (32) along with chain rule, we have

$$\frac{\partial}{\partial\theta}\theta^{i,t}(\theta) = \frac{\partial}{\partial\theta}\left(\Psi_i(...(\Psi_i(\theta, \mathcal{D}^i_{test,1})...), \mathcal{D}^i_{test,t})\right) = \prod^t_{t'=1}(I + \alpha\tilde{\nabla}^2 J_i(\theta^{i,t'-1}(\theta), \mathcal{D}^i_{test,t'})) \tag{33}$$

for any $t \geq 1$.

Using the formulation for derivative of product of functions, we obtain:

$$\nabla V_\zeta(\theta) = \nabla_\theta \mathbb{E}_{i \sim p} \left[ \sum_{\{\mathcal{D}^i_{test,t}\}^\zeta_{t=1}} J_i\left(\theta^{i,\zeta}(\theta)\right) \prod_{t=1}^\zeta q_i(\mathcal{D}^i_{test,t}; \theta^{i,t-1}(\theta)) \right]$$

$$= \mathbb{E}_{i \sim p} \left[ \sum_{\{\mathcal{D}^i_{test,t}\}^\zeta_{t=1}} \frac{\partial}{\partial \theta}\left(J_i\left(\theta^{i,\zeta}(\theta)\right)\right) \prod_{t=1}^\zeta q_i(\mathcal{D}^i_{test,t}; \theta^{i,t-1}(\theta)) \right.$$

$$\left. + \sum_{\{\mathcal{D}^i_{test,t}\}^\zeta_{t=1}} \left( J_i\left(\theta^{i,\zeta}(\theta)\right) \sum_{t=1}^\zeta \left( \frac{\partial}{\partial \theta}\left(q_i(\mathcal{D}^i_{test,t}; \theta^{i,t-1}(\theta))\right) \prod_{\substack{t'=1 \\ t' \neq t}}^\zeta q_i(\mathcal{D}^i_{test,t'}; \theta^{i,t'-1}(\theta)) \right) \right) \right].$$

$$(34)$$

Now, note that, by using chain rule, we have

$$\frac{\partial}{\partial \theta}\left(q_i(\mathcal{D}^i_{test,t}; \theta^{i,t-1}(\theta))\right) = \frac{\partial}{\partial \theta}\theta^{i,t-1}(\theta)\nabla_\theta q_i(\mathcal{D}^i_{test,t}; \theta^{i,t-1}(\theta))$$

$$= \frac{\partial}{\partial \theta}\theta^{i,t-1}(\theta)\nabla_\theta \log q_i(\mathcal{D}^i_{test,t}; \theta^{i,t-1}(\theta))q_i(\mathcal{D}^i_{test,t}; \theta^{i,t-1}(\theta)) \quad (35)$$

Plugging (35) in (34), we obtain

$$\nabla V_\zeta(\theta) =$$

$$= \mathbb{E}_{i \sim p} \left[ \sum_{\{\mathcal{D}^i_{test,t}\}^\zeta_{t=1}} \frac{\partial}{\partial \theta}\left(J_i\left(\theta^{i,\zeta}(\theta)\right)\right) \prod_{t=1}^\zeta q_i(\mathcal{D}^i_{test,t}; \theta^{i,t-1}(\theta)) \right.$$

$$\left. + \sum_{\{\mathcal{D}^i_{test,t}\}^\zeta_{t=1}} \left( J_i\left(\theta^{i,\zeta}(\theta)\right) \sum_{t=1}^\zeta \left( \frac{\partial}{\partial \theta}\theta^{i,t-1}(\theta)\nabla_\theta \log q_i(\mathcal{D}^i_{test,t}; \theta^{i,t-1}(\theta)) \right) \prod_{t=1}^\zeta q_i(\mathcal{D}^i_{test,t}; \theta^{i,t-1}(\theta)) \right) \right]$$

$$= \mathbb{E}_{i \sim p}\mathbb{E}_{\{\mathcal{D}^i_{test,j}\}^\zeta_{t=1}} \left[ \frac{\partial}{\partial \theta}\left(J_i\left(\theta^{i,\zeta}(\theta)\right)\right) + J_i\left(\theta^{i,\zeta}(\theta)\right)\sum_{t=1}^\zeta \left( \frac{\partial}{\partial \theta}\theta^{i,t-1}(\theta)\nabla_\theta \log q_i(\mathcal{D}^i_{test,t}; \theta^{i,t-1}(\theta)) \right) \right]$$

$$= \mathbb{E}_{i \sim p}\mathbb{E}_{\{\mathcal{D}^i_{test,j}\}^\zeta_{t=1}} \left[ \frac{\partial}{\partial \theta}\theta^{i,\zeta}(\theta)\nabla J_i(\theta^{i,\zeta}(\theta)) \right.$$

$$\left. + J_i\left(\theta^{i,\zeta}(\theta)\right)\sum_{t=1}^\zeta \left( \frac{\partial}{\partial \theta}\theta^{i,t-1}(\theta)\nabla_\theta \log q_i(\mathcal{D}^i_{test,t}; \theta^{i,t-1}(\theta)) \right) \right] \quad (36)$$

where the last equality is derived by substituting $\frac{\partial}{\partial \theta}\left(J_i\left(\theta^{i,\zeta}(\theta)\right)\right)$ by $\frac{\partial}{\partial \theta}\theta^{i,t-1}(\theta)\nabla J_i(\theta^{i,\zeta}(\theta))$ by using chain rule. Now, we characterize $\nabla_\theta \log q_i(\mathcal{D}^i_{test,t}; \theta^{i,t-1}(\theta))$ which appears in (36). First, recall that

$$\nabla_\theta \log q_i(\mathcal{D}^i_{test,t}; \theta^{i,t-1}(\theta)) = \sum_{\tau \in \mathcal{D}^i_{test,t}} \nabla_\theta \log q_i(\tau; \theta^{i,t-1}(\theta)).$$

Therefore,

$$\nabla_\theta \log q_i(\mathcal{D}^i_{test,t}; \theta^{i,t-1}(\theta)) = \sum_{\tau \in \mathcal{D}^i_{test,t}} \nabla_\theta \log q_i(\tau; \theta^{i,t-1}(\theta))$$

$$= \sum_{\tau=((s_j,a_j)^H_{j=0}) \in \mathcal{D}^i_{test,t}} \sum_{h=0}^H \nabla_\theta \log \pi_i(a_h|s_h; \theta^{i,t-1}(\theta))$$

$$= \sum_{\tau \in \mathcal{D}^i_{test,t}} \nabla_\theta \log \pi_i(\tau; \theta^{i,t-1}(\theta)) \quad (37)$$

**Input:** Initial iterate $\theta_0$
**repeat**
    Draw a batch of *i.i.d.* tasks (MDPs) $\mathcal{B}_k \subseteq \mathcal{I}$ from distribution $p$ and with size $B = |\mathcal{B}_k|$;
    Set $\theta_{k+1}^{i,0} = \theta_k$;
    **for** all $\mathcal{T}_i$ with $i \in \mathcal{B}_k$ **do**
        **for** $t \leftarrow 1$ to $\zeta$ **do**
            Sample a batch of trajectories $\mathcal{D}_{in,t}^i$ w.r.t. $q_i(.; \theta_{k+1}^{i,t-1})$;
            Set $\theta_{k+1}^{i,t} = \theta_{k+1}^{i,t-1} + \alpha \tilde{\nabla} J_i(\theta_{k+1}^{i,t-1}, \mathcal{D}_{in,t}^i)$;
        **end for**
    **end for**
    Set $\theta_{k+1} = \theta_k + \beta \tilde{\nabla} V_\zeta(\theta_k; \mathcal{B}_k, \{\mathcal{D}_{in,t}^i\}_{i,t}, \mathcal{D}_o^i)$ where $\tilde{\nabla} V_\zeta(.;.)$ is given by (39);
    $k \leftarrow k + 1$
**until** not done

where the second equality follows from (1) and we used the notation (11) for the last equality. Plugging (37) and (33) in (36), we obtain

$$
\nabla V_\zeta(\theta) = \mathbb{E}_{i \sim p} \mathbb{E}_{\{\mathcal{D}_{test,j}^i\}_{t=1}^\zeta} \left[ \prod_{t=1}^\zeta (I + \alpha \tilde{\nabla}^2 J_i(\theta^{i,t-1}(\theta), \mathcal{D}_{test,t'}^i)) \nabla J_i(\theta^{i,\zeta}(\theta)) \right.
$$
$$
\left. + J_i\left(\theta^{i,\zeta}(\theta)\right) \sum_{t=1}^\zeta \left( \prod_{t'=1}^{t-1} (I + \alpha \tilde{\nabla}^2 J_i(\theta^{i,t'-1}(\theta), \mathcal{D}_{test,t'}^i)) \sum_{\tau \in \mathcal{D}_{test,t}^i} \nabla_\theta \log \pi_i(\tau; \theta^{i,t-1}(\theta)) \right) \right].
$$
$$(38)$$

$\square$

As a consequence,

$$
\tilde{\nabla} V_\zeta(\theta; \mathcal{B}_k, \{\mathcal{D}_{in,t}^i\}_{i,t}, \mathcal{D}_o^i) := \frac{1}{B} \sum_{i \in \mathcal{B}_k} \left( \prod_{t=1}^\zeta (I + \alpha \tilde{\nabla}^2 J_i(\theta^{i,t-1}(\theta), \mathcal{D}_{in,t'}^i)) \tilde{\nabla} J_i(\theta^{i,\zeta}(\theta), \mathcal{D}_o^i) \right.
$$
$$
\left. + \tilde{J}_i\left(\theta^{i,\zeta}(\theta), \mathcal{D}_o^i\right) \sum_{t=1}^\zeta \left( \prod_{t'=1}^{t-1} (I + \alpha \tilde{\nabla}^2 J_i(\theta^{i,t'-1}(\theta), \mathcal{D}_{in,t'}^i)) \sum_{\tau \in \mathcal{D}_{in,t}^i} \nabla_\theta \log \pi_i(\tau; \theta^{i,t-1}(\theta)) \right) \right)
$$
$$(39)$$

is an unbiased estimate of $\nabla V_\zeta(\theta)$ where $\mathcal{B}_k$ is a batch of tasks drawn independently from distribution $p$ and $\mathcal{D}_{in,t}^i$ and $\mathcal{D}_o^i$ are batch of trajectories drawn according to $q_i(.; \theta_{k+1}^{i,t-1})$ and $q_i(.; \theta_{k+1}^{i,\zeta})$, respectively. The steps of SG-MRL using this unbiased estimate are illustrated in Algorithm 2.

## D   On Softmax Policy

First, we show that

$$
\nabla_\theta^2 \log \pi(a|s; \theta) =
$$
$$
- \mathbb{E}_{a' \sim \pi(a'|s,\theta)} \left[ \left( \phi(a', s) - \mathbb{E}_{a'' \sim \pi(a''|s,\theta)}[\phi(a'', s)] \right) \left( \phi(a'', s) - \mathbb{E}_{a'' \sim \pi(a'|s,\theta)}[\phi(a'', s)] \right)^\top \right].
$$
$$(40)$$

Note that

$$\nabla_\theta^2 \log \pi(a|s;\theta) = -\frac{\partial}{\partial\theta}\mathbb{E}_{a'\sim\pi(a'|s,\theta)}[\phi(a',s)]$$

$$= -\frac{\partial}{\partial\theta}\sum_{a'\in\mathcal{A}}\pi(a'|s,\theta)\phi(a',s)$$

$$= -\sum_{a'\in\mathcal{A}}\phi(a',s)\nabla_\theta\pi(a'|s,\theta)^\top \tag{41}$$

$$= -\sum_{a'\in\mathcal{A}}\phi(a',s)\nabla_\theta\log\pi(a'|s,\theta)^\top\pi(a'|s,\theta) \tag{42}$$

$$= -\mathbb{E}_{a'\sim\pi(a'|s,\theta)}\left[\phi(a',s)\nabla_\theta\log\pi(a'|s,\theta)^\top\right]$$

$$= -\mathbb{E}_{a'\sim\pi(a'|s,\theta)}\left[\phi(a',s)\left(\phi(a',s) - \mathbb{E}_{a''\sim\pi(a''|s,\theta)}[\phi(a'',s)]\right)^\top\right] \tag{43}$$

$$= -\mathbb{E}_{a'\sim\pi(a'|s,\theta)}\left[\phi(a',s)\phi(a',s)^\top\right] + \mathbb{E}_{a'\sim\pi(a'|s,\theta)}[\phi(a',s)](\mathbb{E}_{a'\sim\pi(a'|s,\theta)}[\phi(a',s)])^\top$$

where (42) follows from the log trick, i.e., the fact that $\nabla_\theta\pi(a'|s,\theta) = \nabla_\theta\log\pi(a'|s,\theta)\pi(a'|s,\theta)$, and (43) is obtained using (29).

Next, we assume $\phi(.,.)$ is bounded and want to show $\nabla_\theta^2\log\pi(a|s;\theta)$ is a Lipschitz function of $\theta$. First, note that $\nabla_\theta\log\pi(a|s;\theta)$ given by (29) is bounded due to boundedness of $\phi(.,.)$. Thus, by Lemma 3, $\mathbb{E}_{a''\sim\pi(a''|s,\theta)}[\phi(a',s)]$ is Lipschitz, and it is also bounded as $\phi(.,.)$ is bounded. Hence, the term

$$\left(\phi(a',s) - \mathbb{E}_{a''\sim\pi(a''|s,\theta)}[\phi(a'',s)]\right)\left(\phi(a'',s) - \mathbb{E}_{a''\sim\pi(a'|s,\theta)}[\phi(a'',s)]\right)^\top$$

is bounded, as it is also Lipschitz by Lemma 2. Finally, applying Lemma 3 one more time shows (40) is Lipschitz which completes the proof.

## E    Proof of Lemma 1

**Proof of (1) & (2)**: check [22].

**Proof of (3):** Note that it suffices to show the for one trajectory $\tau$, $u_i(\tau;\theta)$ is Lipschitz with parameter $\eta_\rho$ as

$$\|\tilde\nabla^2 J_i(\theta_1,\mathcal{D}^i) - \tilde\nabla^2 J_i(\theta_2,\mathcal{D}^i)\| \le \frac{1}{|\mathcal{D}^i|}\sum_{\tau\in\mathcal{D}^i}\|u_i(\tau;\theta_1) - u_i(\tau;\theta_2)\|. \tag{44}$$

Let $\tau = (s_0, a_0, ..., s_H, a_H)$. Recall that

$$u_i(\tau;\theta) = g_i(\tau;\theta)\nabla_\theta\log q_i(\tau;\theta)^\top + \nabla_\theta^2\nu_i(\tau;\theta)$$

$$= g_i(\tau;\theta)\left(\sum_{h=0}^H\nabla_\theta\log\pi_i(a_h|s_h;\theta)\right)^\top + \sum_{h=0}^H\nabla^2\log\pi_i(a_h|s_h;\theta)\mathcal{R}_i^h(\tau). \tag{45}$$

We now show both terms in (45) are Lipschitz and characterize their Lipschitz parameters. First, note that $g_i(\tau;\theta)$ is bounded by $\eta_G$. Also, note that

$$\|g_i(\tau;\theta_1) - g_i(\tau;\theta_2)\| = \|\sum_{h=0}^H\left((\nabla_\theta\log\pi_i(a_h|s_h;\theta_1) - \nabla_\theta\log\pi_i(a_h|s_h;\theta_2))\mathcal{R}_i^h(\tau)\right)\|$$

$$\le \sum_{h=0}^H\left(\|\nabla_\theta\log\pi_i(a_h|s_h;\theta_1) - \nabla_\theta\log\pi_i(a_h|s_h;\theta_2)\|\mathcal{R}_i^h(\tau)\right)$$

$$\le \sum_{h=0}^H\left(L\|\theta_1 - \theta_2\|\mathcal{R}_i^h(\tau)\right) \tag{46}$$

$$\le L\|\theta_1 - \theta_2\|\sum_{h=0}^H\frac{R\gamma^h}{1-\gamma} \tag{47}$$

$$\le \frac{LR}{(1-\gamma)^2}\|\theta_1 - \theta_2\|$$

where (46) follows from Assumption 2 and (47) is obtained using the fact that $\mathcal{R}_i^h(\tau) \leq \frac{R\gamma^h}{1-\gamma}$. In addition, $\sum_{h=0}^{H} \nabla_\theta \log \pi_i(a_h|s_h;\theta)$ is bounded by $(H+1)G$ and is Lipschitz with parameter $(H+1)L$ due to Assumption 2. As a result, by Lemma 2, the first term of (45), i.e., $g_i(\tau;\theta) \left( \sum_{h=0}^{H} \nabla_\theta \log \pi_i(a_h|s_h;\theta) \right)^\top$ is Lipschitz with parameter $\eta_G(H+1)L + (H+1)G\frac{LR}{(1-\gamma)^2}$. Replacing $\eta_G$ implies that Lipschitz parameter is in fact $2(H+1)GLR/(1-\gamma)^2$.

For the second term of (45), note that using Assumption 3 yields

$$\left\| \sum_{h=0}^{H} \left( (\nabla^2 \log \pi_i(a_h|s_h;\theta) - \nabla^2 \log \pi_i(a_h|s_h;\theta))\mathcal{R}_i^h(\tau) \right) \right\| \leq \sum_{h=0}^{H} \left( \rho\|\theta_1 - \theta_2\|\mathcal{R}_i^h(\tau) \right)$$

$$\leq \rho\|\theta_1 - \theta_2\| \sum_{h=0}^{H} \frac{R\gamma^h}{1-\gamma} \leq \frac{\rho R}{(1-\gamma)^2}\|\theta_1 - \theta_2\|$$

where the second inequality once again follows from $\mathcal{R}_i^h(\tau) \leq \frac{R\gamma^h}{1-\gamma}$. Adding up the Lipschitz parameters of both terms of (45) completes the proof.

## F  On Boundedness and Lipschitz Property of $\nabla V_\zeta(\theta)$

In the following Theorem, we characterize boundedness and Lipschitz property of $\nabla V_\zeta(\theta)$ for any $\zeta \geq 1$.

**Theorem 3.** *Consider the objective function $V_\zeta$ defined in (7) for the case that $\alpha \in (0, 1/\eta_H]$ where $\eta_H$ is given in Lemma 1. Suppose that the conditions in Assumptions 1-3 are satisfied. Then, for any $\theta \in \mathbb{R}^d$, the norm of $\nabla V_\zeta(\theta)$ is upper bounded by*

$$G_V(\zeta) := 2^\zeta(\eta_G + D_{in}GR(H+1)) = 2^\zeta GR\left( \frac{1}{(1-\gamma)^2} + D_{in}(H+1) \right). \tag{48}$$

*Moreover, $\nabla V_\zeta(\theta)$ is Lipschitz with parameter*

$$L_V(\zeta) := \zeta 2^{\zeta-1}\alpha\eta_\rho\eta_G + 2^{2\zeta}\eta_H \tag{49}$$
$$+ 2^\zeta D_{in}(H+1)\left( R\left( 2^\zeta L + (\zeta + 2^\zeta)D_{in}G^2(H+1) + (\zeta - 1)\alpha\eta_\rho G \right) + 2^{\zeta+1}\eta_G G \right)$$

*where $\eta_G$ and $\eta_\rho$ are also defined in Lemma 1.*

*Proof.* Recall from (36) in Appendix C that

$$\nabla V_\zeta(\theta) = \mathbb{E}_{i\sim p}\mathbb{E}_{\{\mathcal{D}_{test,j}^i\}_{t=1}^\zeta}\left[ \frac{\partial}{\partial\theta}\theta^{i,\zeta}(\theta)\nabla J_i(\theta^{i,\zeta}(\theta)) \right.$$

$$\left. + J_i\left( \theta^{i,\zeta}(\theta) \right) \sum_{t=1}^\zeta \left( \frac{\partial}{\partial\theta}\theta^{i,t-1}(\theta)\nabla_\theta \log q_i(\mathcal{D}_{test,t}^i; \theta^{i,t-1}(\theta)) \right) \right]$$

$$= \mathbb{E}_{i\sim p}\left[ \sum_{\{\mathcal{D}_{test,t}\}_{t=0}^\zeta} \left( \prod_{t=1}^\zeta q_i(\mathcal{D}_{test,t}^i; \theta^{i,t-1}(\theta)) \left( \frac{\partial}{\partial\theta}\theta^{i,\zeta}(\theta)\nabla J_i(\theta^{i,\zeta}(\theta)) \right. \right. \right.$$

$$\left. \left. \left. + J_i\left( \theta^{i,\zeta}(\theta) \right) \sum_{t=1}^\zeta \left( \frac{\partial}{\partial\theta}\theta^{i,t-1}(\theta)\nabla_\theta \log q_i(\mathcal{D}_{test,t}^i; \theta^{i,t-1}(\theta)) \right) \right) \right) \right] \tag{50}$$

where $\theta^{i,0}(\theta) := \theta$ and $\theta^{i,t}(\theta) := \Psi_i(...(\Psi_i(\theta, \mathcal{D}_{test,1}^i)...), \mathcal{D}_{test,t}^i)$ for $t \geq 1$. To show the desired result, we first characterize the boundedness and Lipschitz property of

$$\frac{\partial}{\partial\theta}\theta^{i,\zeta}(\theta)\nabla J_i(\theta^{i,\zeta}(\theta)) + J_i\left( \theta^{i,\zeta}(\theta) \right) \sum_{t=1}^\zeta \left( \frac{\partial}{\partial\theta}\theta^{i,t-1}(\theta)\nabla_\theta \log q_i(\mathcal{D}_{test,t}^i; \theta^{i,t-1}(\theta)) \right) \tag{51}$$

for any $i$ and any sequence of batches $\{\mathcal{D}_{test,t}\}_{t=0}^\zeta$. In particular, we show (51) is bounded by $G_V(\zeta)$, and therefore, the bound holds for $\nabla V_\zeta(\theta)$ as well. Furthermore, we show a bound on the Lipschitz

parameter of (51) which is independent of both $\{\mathcal{D}_{test,t}\}_{t=0}^{\zeta}$ and $i$, and we obtain it by showing each term in (51) is bounded and Lipschitz and then applying Lemma 2. Finally, to show (49), we use Lemma 3.

We now start with studying boundedness and Lipschitz property of (51). In this regard, first, we show the following lemma on the Lipschitz property of $\theta^{i,t}(\theta)$ and its derivative for any $t$:

**Lemma 4.** *Let $t \geq 1$, and recall that $\theta^{i,t}(\theta) := \Psi_i(...(\Psi_i(\theta, \mathcal{D}_{test,1}^i)...), \mathcal{D}_{test,t}^i)$ for a sequence of batch of trajectories $\{D_{test,j}^i\}_{j=1}^t$. Then, for any $\theta, \tilde{\theta}$, we have*

    *1.*

$$\|\frac{\partial}{\partial \theta}\theta^{i,t}(\theta)\| \leq (1 + \alpha\eta_H)^t, \quad \text{and thus } \|\theta^{i,t}(\theta) - \theta^{i,t}(\tilde{\theta})\| \leq (1 + \alpha\eta_H)^t\|\theta - \tilde{\theta}\|, \quad (52)$$

    *2.*

$$\|\frac{\partial}{\partial \theta}\theta^{i,t}(\theta) - \frac{\partial}{\partial \theta}\theta^{i,t}(\tilde{\theta})\| \leq t\alpha\eta_\rho(1 + \alpha\eta_H)^{t-1}\|\theta - \tilde{\theta}\| \quad (53)$$

*where $\eta_H$ and $\eta_\rho$ are given in Lemma 1.*

*Proof.* Recall from (33) in Appendix C that

$$\frac{\partial}{\partial \theta}\theta^{i,t}(\theta) = \prod_{t'=1}^{t}(I + \alpha\tilde{\nabla}^2 J_i(\theta^{i,t'-1}(\theta), \mathcal{D}_{test,t'}^i)) \quad (54)$$

In part (2) of Lemma 1 we showed that for any $t'$, $\|\tilde{\nabla}^2 J_i(\theta^{i,t'-1}(\theta), \mathcal{D}_{test,t'}^i)\| \leq \eta_H$, and this immediately implies the first result.

Also, for the second result, note that for each $t'$, $I + \alpha\tilde{\nabla}^2 J_i(\theta^{i,t'-1}(\theta), \mathcal{D}_{test,t'}^i)$ is bounded by $1 + \alpha\eta_H$ due to part (2) of Lemma 1, and is Lipschitz with parameter $\alpha\eta_\rho$ by part (3) of Lemma 1. Thus, using Lemma 2 gives us the desired result. $\square$

Next, we go step by step and study the boundedness and Lipschitz property of each term in (51). Throughout this process, we also use the assumption $\alpha \leq 1/\eta_H$ to replace the term $(1 + \alpha\eta_H)$ by 2 and simplify the results.

    (i) As we showed in Lemma 4, $\frac{\partial}{\partial \theta}\theta^{i,\zeta}(\theta)$ is bounded by $2^\zeta$ and also Lipschitz with parameter $\zeta\alpha\eta_\rho 2^{\zeta-1}$. Also, $\nabla J_i(\theta^{i,\zeta}(\theta))$ is bounded by $\eta_G$ by part (1) of Lemma 1 and is Lipschitz with parameter $\eta_H 2^\zeta$ by using part (2) of Lemma 1 and Lemma 4 along with the fact that the Lipschitz parameter of combination of functions is the product of their Lipschitz parameters. Thus, using Lemma 2, the term $\frac{\partial}{\partial \theta}\theta^{i,\zeta}(\theta)\nabla J_i(\theta^{i,\zeta}(\theta))$ in total is bounded by $\eta_G 2^\zeta$ and is Lipschitz with parameter $\zeta 2^{\zeta-1}\alpha\eta_\rho\eta_G + 2^{2\zeta}\eta_H$.

    (ii) For any $t$, and by Lemma 4, $\frac{\partial}{\partial \theta}\theta^{i,t-1}(\theta)$ is bounded by $2^{t-1}$ and its Lipschitz parameter is bounded by $(t-1)2^{t-1}\alpha\eta_\rho$.

      Also, it is easy to check

$$\|\nabla_\theta \log q_i(\mathcal{D}_{test,t}^i; \theta)\| \leq D_{in}G(H+1), \quad \|\nabla_\theta^2 \log q_i(\mathcal{D}_{test,t}^i; \theta)\| \leq D_{in}L(H+1). \quad (55)$$

      Hence, $\nabla_\theta \log q_i(\mathcal{D}_{test,t}^i; \theta^{i,t-1}(\theta))$ is bounded by $D_{in}G(H + 1)$. In addition, since $\theta^{i,t-1}(\theta)$ is Lipschitz with parameter $2^{t-1}$, the whole $\nabla_\theta \log q_i(\mathcal{D}_{test,t}^i; \theta^{i,t-1}(\theta))$ is Lipschitz with parameter $2^{t-1}D_{in}L(H + 1)$.

      Thus, for any $t$, the term $\frac{\partial}{\partial \theta}\theta^{i,t-1}(\theta)\nabla_\theta \log q_i(\mathcal{D}_{test,t}^i; \theta^{i,t-1}(\theta))$ is bounded by $2^{t-1}D_{in}G(H+1)$ and is Lipschitz with parameter $D_{in}(H+1)(2^{2t-2}L+(t-1)2^{t-1}\alpha\eta_\rho G)$. As a consequence, the sum

$$\sum_{t=1}^{\zeta}\left(\frac{\partial}{\partial \theta}\theta^{i,t-1}(\theta)\nabla_\theta \log q_i(\mathcal{D}_{test,t}^i; \theta^{i,t-1}(\theta))\right) \quad (56)$$

is bounded by $2^\zeta D_{in}G(H+1)$ and its Lipschitz parameter is bounded by

$$D_{in}(H+1)\left(4^\zeta L + 2^\zeta(\zeta-1)\alpha\eta_\rho G\right).$$

(iii) $J_i\left(\theta^{i,\zeta}(\theta)\right)$ is clearly bounded by $R$. Also, by part(1) of Lemma 1 $J_i$ is Lipschitz with parameter $\eta_G$ and also by Lemma 4, $\theta^{i,\zeta}(\theta)$ is Lipschitz with parameter $2^\zeta$. Using these two along with the fact that Lipschitz parameter of combination of functions is equal to the product of their Lipschitz parameters, implies that $J_i\left(\theta^{i,\zeta}(\theta)\right)$ is Lipschitz with parameter $2^\zeta\eta_G$.

(iv) Therefore, using (iv) and (v), the whole term

$$\prod_{t=1}^\zeta q_i(\mathcal{D}^i_{test,t}; \theta^{i,t-1}(\theta))J_i\left(\theta^{i,\zeta}(\theta)\right)\sum_{t=1}^\zeta\left(\frac{\partial}{\partial\theta}\theta^{i,t-1}(\theta)\nabla_\theta\log q_i(\mathcal{D}^i_{test,t}; \theta^{i,t-1}(\theta))\right) \tag{57}$$

is bounded by $2^\zeta D_{in}GR(H+1)$ and, by Lemma 2, its Lipschitz parameter is bounded by

$$D_{in}R(H+1)\left(4^\zeta L + 2^\zeta(\zeta-1)\alpha\eta_\rho G\right) + 2^{2\zeta}D_{in}G(H+1)\eta_G + R\zeta 2^\zeta D_{in}^2 G^2(H+1)^2.$$

which can be simplified and written as

$$2^\zeta D_{in}(H+1)\left(R\left(2^\zeta L + \zeta D_{in}G^2(H+1) + (\zeta-1)\alpha\eta_\rho G\right) + 2^\zeta\eta_G G\right)$$

Part (i) and (iv) together imply that (51) is bounded by

$$2^\zeta(\eta_G + D_{in}GR(H+1)) = 2^\zeta GR\left(\frac{1}{(1-\gamma)^2} + D_{in}(H+1)\right) \tag{58}$$

which is in fact $G_V(\zeta)$. Since this upper bound is independent of $i$ and $\{D^i_{test,t}\}_t$, it also holds for $\nabla V_\zeta(\theta)$, and this completes the proof of (48).

Also, part (i) and (iv) together imply that (51) is Lipschitz with parameter

$$\zeta 2^{\zeta-1}\alpha\eta_\rho\eta_G + 2^{2\zeta}\eta_H + 2^\zeta D_{in}(H+1)\left(R\left(2^\zeta L + \zeta D_{in}G^2(H+1) + (\zeta-1)\alpha\eta_\rho G\right) + 2^\zeta\eta_G G\right). \tag{59}$$

Now, to derive the Lipschitz parameter of $\nabla V_\zeta(\theta)$ itself, we use Lemma 3. To do so, first we show the following lemma.

**Lemma 5.** *Recall definition of $q_i(\mathcal{D}^i; \theta)$ (18) for some MDP $\mathcal{M}_i$, batch of trajectories $\mathcal{D}^i$ and policy parameter $\theta \in \mathbb{R}^d$. Then, for any $\mathcal{D}^i$ and $\theta$, we have*

$$\|\nabla_\theta\log q_i(\mathcal{D}^i; \theta)\| \le |\mathcal{D}^i|(H+1)G. \tag{60}$$

*Proof.* Note that

$$\|\nabla_\theta\log q_i(\mathcal{D}^i; \theta)\| = \left\|\sum_{\tau\in\mathcal{D}^i}\nabla_\theta\log\pi_i(\tau; \theta)\right\| \tag{61}$$

$$\le |D^i|\max_{\tau=(s_0,a_0,\ldots,s_H,a_H)}\|\nabla_\theta\log\pi_i(\tau; \theta)\|$$

$$\le |D^i|\max_{\tau=(s_0,a_0,\ldots,s_H,a_H)}\sum_{h=0}^H\|\nabla_\theta\log\pi_i(a_h|s_h; \theta)\| \tag{62}$$

$$\le |\mathcal{D}^i|(H+1)G \tag{63}$$

where (61) follows from (18) and (62) is obtained using (11) along with Assumption 2. $\qquad\square$

Using this lemma, we have

$$\|\nabla_\theta \left( \log \prod_{t=1}^\zeta q_i(\mathcal{D}_{test,t}^i; \theta^{i,t-1}(\theta)) \right) \| \le \sum_{t=1}^\zeta \| \frac{\partial}{\partial \theta} \theta^{i,t-1}(\theta) \nabla_\theta \log q_i(\mathcal{D}_{test,t}^i; \theta^{i,t-1}(\theta)) \|$$

$$\le |D_{in}|(H+1)G \sum_{t=1}^\zeta \| \frac{\partial}{\partial \theta} \theta^{i,t-1}(\theta) \| \tag{64}$$

$$\le |D_{in}|(H+1)G \sum_{t=1}^\zeta 2^{t-1} \tag{65}$$

$$\le 2^\zeta |D_{in}|(H+1)G$$

where (64) follows from Lemma 5 and (65) is obtained using Lemma 4. Now, using this bound and (59) along with Lemma 3 implies that $\nabla V_\zeta(\theta)$ is Lipschitz with parameter

$$\zeta 2^{\zeta-1} \alpha \eta_\rho \eta_G + 2^{2\zeta} \eta_H + 2^\zeta D_{in}(H+1) \left( R \left( 2^\zeta L + (\zeta + 2^\zeta) D_{in} G^2(H+1) + (\zeta-1)\alpha \eta_\rho G \right) + 2^{\zeta+1} \eta_G G \right) \tag{66}$$

which completes the proof of (49). $\qquad\square$

In particular, for $\zeta = 1$, it is easy to verify the Lipschitz parameter of $\nabla V_1(\theta)$ admits the upper bound

$$\alpha \eta_\rho \eta_G + 4\eta_H + 8RD_{in}(H+1)(L + D_{in}G^2(H+1)). \tag{67}$$

Finally, we state the following result on boundedness of unbiased estimate of $\nabla V_\zeta(\theta)$ used in update of MAML (Algorithm 2).

**Lemma 6.** *Recall* $\tilde{\nabla} V_\zeta(\theta_k; \mathcal{B}_k, \{\mathcal{D}_{in,t}^i\}_{i,t}, \mathcal{D}_o^i)$ (39) *in Multi-step MAML algorithm (Algorithm 2) for the case that* $\alpha \in (0, 1/\eta_H]$ *where* $\eta_H$ *is given in Lemma 1. Suppose that the conditions in Assumptions 1-3 are satisfied. Then, at iteration* $k+1$, *and for any choice of* $\mathcal{B}_k$, $\{\mathcal{D}_o^i\}_i$ *and* $\{\mathcal{D}_{in,t}^i\}_{i,t}$, *we have*

$$\|\tilde{\nabla} V_\zeta(\theta_k; \mathcal{B}_k, \{\mathcal{D}_{in,t}^i\}_{i,t}, \mathcal{D}_o^i)\| \le G_V(\zeta) \tag{68}$$

*where* $G_V(\zeta)$ *is given in Theorem 3.*

*Proof.* We skip the details of the proof as it can be done very similar to how we proved (51) in Theorem 3. In particular, note that for any choice of $\mathcal{D}_o^i$

$$\|\tilde{\nabla} J_i(\theta^{i,\zeta}(\theta), \mathcal{D}_o^i)\| \le \eta_G, \quad \|\tilde{J}_i(\theta^{i,\zeta}(\theta), \mathcal{D}_o^i)\| \le R \tag{69}$$

where the first one follows from Lemma 1 and the second one is an immediate result of Assumption 1. $\qquad\square$

## G Proof of Theorem 1

We first state the general statement of the theorem for any $\zeta \ge 1$.

**Theorem 4.** *Consider the objective function* $V_\zeta$ *defined in (7) for the case that* $\alpha \in (0, 1/\eta_H]$ *where* $\eta_H$ *is given in Lemma 1. Suppose that the conditions in Assumptions 1-3 are satisfied, and recall the definitions* $L_V(\zeta)$ *and* $G_V(\zeta)$ *from Theorem 3. Consider running Multi-step SG-MRL (Algorithm 2) with* $\beta \in (0, 1/L_V(\zeta)]$. *Then, for any* $1 > \epsilon > 0$, *MAML finds a solution* $\theta_\epsilon$ *such that*

$$\mathbb{E}[\|\nabla V_\zeta(\theta_\epsilon)\|^2] \le \frac{2G_V(\zeta)^2 L_V(\zeta)\beta}{BD_o} + \epsilon^2 \tag{70}$$

*after at most running for*

$$\mathcal{O}(1) \frac{R}{\beta} \min \left\{ \frac{1}{\epsilon^2}, \frac{BD_o}{G_V(\zeta)^2 L_V(\zeta)\beta} \right\} \tag{71}$$

*iterations.*

*Proof.* Throughout the proof, we use $G_V$ and $L_V$ instead of $G_V(\zeta)$ and $L_V(\zeta)$, respectively, to simplify the notation. Also, we denote the filtration till the end of iteration $k$ by $\mathcal{F}_k$.

As we previously discussed, $\tilde{\nabla}V_\zeta(\theta_k; \mathcal{B}_k, \{\mathcal{D}^i_{in,t}\}_{i,t}, \mathcal{D}^i_o)$ is an unbiased estimate of $\nabla V_\zeta(\theta_k)$ at iteration $k+1$. In the following lemma, we upper bound the variance of this estimation.

**Lemma 7.** *Recall the definition of $\tilde{\nabla}V_\zeta(\theta_k; \mathcal{B}_k, \{\mathcal{D}^i_{in,t}\}_{i,t}, \mathcal{D}^i_o)$ (39) in Multi-step SG-MRL algorithm (Algorithm 2) for the case that $\alpha \in (0, 1/\eta_H]$ where $\eta_H$ is given in Lemma 1. Suppose that the conditions in Assumptions 1-3 are satisfied. Then, at iteration $k+1$, and for any choice of $\mathcal{B}_k$, $\{\mathcal{D}^i_o\}_i$ and $\{\mathcal{D}^i_{in,t}\}_{i,t}$, we have*

$$\mathbb{E}\left[\left\|\tilde{\nabla}V_\zeta(\theta_k; \mathcal{B}_k, \{\mathcal{D}^i_{in,t}\}_{i,t}, \mathcal{D}^i_o) - \nabla V_\zeta(\theta_k)\right\|^2\right] \leq \frac{G_V^2}{BD_o} \tag{72}$$

*where $G_V$ is given in Theorem 3.*

*Proof.* Note that

$$\tilde{\nabla}V_\zeta(\theta_k; \mathcal{B}_k, \{\mathcal{D}^i_{in,t}\}_{i,t}, \mathcal{D}^i_o) = \frac{1}{BD_o}\sum_{i\in\mathcal{B}_k}\sum_{\tau\in\mathcal{D}^i_o}\tilde{\nabla}V_\zeta(\theta_k; \{i\}, \{\mathcal{D}^i_{in,t}\}_{i,t}, \{\tau\}), \tag{73}$$

where for any $i$ and $\tau \in \mathcal{D}^i_o$, $\tilde{\nabla}V_\zeta(\theta_k; \{i\}, \{\mathcal{D}^i_{in,t}\}_{i,t}, \{\tau\})$ is an unbiased estimate of $\nabla V_\zeta(\theta_k)$, and by Lemma 6, its second moment is bounded by $G_V^2$. Also, note that $\tilde{\nabla}V_\zeta(\theta_k; \{i\}, \{\mathcal{D}^i_{in,t}\}_{i,t}, \{\tau\})$ are independent for different $i$ and $\tau$. Finally, to complete the proof, we use the well-known fact that if $\{X_i\}_{i=1}^n$ are independent with mean $\mu$, and for each $i$, variance of $X_i$ is upper bounded by $\sigma^2$, then

$$\mathbb{E}\left[\left\|\frac{X_1 + ... + X_n}{n} - \mu\right\|^2\right] \leq \frac{\sigma^2}{n}.$$

$\square$

Now, we get back to the proof of the main result. From now, and to simplify the notation, we use $\tilde{\nabla}V_\zeta(\theta_k)$ to denote $\tilde{\nabla}V_\zeta(\theta_k; \mathcal{B}_k, \{\mathcal{D}^i_{in,t}\}_{i,t}, \mathcal{D}^i_o)$. Next, note that, using the smoothness property of $\nabla V_\zeta(\theta)$, we have [30]

$$\left|V_\zeta(\theta_{k+1}) - V_\zeta(\theta_k) - \nabla V_\zeta(\theta_k)^\top(\theta_{k+1} - \theta_k)\right| \leq \frac{L_V^2}{2}\|\theta_{k+1} - \theta_k\|^2. \tag{74}$$

Recall that, at iteration $k+1$, MAML performs

$$\theta_{k+1} = \theta_k + \beta\tilde{\nabla}V_\zeta(\theta_k). \tag{75}$$

Plugging this in (74), we obtain

$$-V_\zeta(\theta_{k+1}) \leq -V_\zeta(\theta_k) - \nabla V_\zeta(\theta_k)^\top(\theta_{k+1} - \theta_k) + \frac{L_V^2}{2}\|\theta_{k+1} - \theta_k\|^2$$

$$= -V_\zeta(\theta_k) - \beta\nabla V_\zeta(\theta_k)^\top\tilde{\nabla}V_\zeta(\theta_k) + \frac{L_V^2}{2}\beta^2\|\tilde{\nabla}V_\zeta(\theta_k)\|^2 \tag{76}$$

where the last equality follows from (75). Next, taking expectation from both sides and conditioning on $\mathcal{F}_k$, implies

$$-\mathbb{E}[V_\zeta(\theta_{k+1})|\mathcal{F}_k]$$
$$\leq -V_\zeta(\theta_k) - \beta\|\nabla V_\zeta(\theta_k)\|^2 + \frac{L_V}{2}\beta^2\left(\|\nabla V_\zeta(\theta_k)\|^2 + \mathbb{E}\left[\|\tilde{\nabla}V_\zeta(\theta_k) - \nabla V_\zeta(\theta_k)\|^2|\mathcal{F}_k\right]\right) \tag{77}$$

$$\leq -V_\zeta(\theta_k) - \frac{\beta}{2}\|\nabla V_\zeta(\theta_k)\|^2 + \frac{G_V^2 L_V \beta^2}{2BD_o} \tag{78}$$

where the first inequality is obtained using the fact that $\tilde{\nabla}V_\zeta(\theta_k)$ is an unbiased estimate of $\nabla V_\zeta(\theta_k)$ and $\nabla V_\zeta(\theta_k)$ is deterministic condition on $\mathcal{F}_k$. (78) is also an immediate result of Lemma 7 along with $\beta \leq 1/L_V$.

Taking another expectation from both sided of (78), and using tower rule, we obtain

$$-\mathbb{E}[V_\zeta(\theta_{k+1})] \leq -\mathbb{E}[V_\zeta(\theta_k)] - \frac{\beta}{2}\mathbb{E}\left[\|\nabla V_\zeta(\theta_k)\|^2\right] + \frac{G_V^2 L_V \beta^2}{2BD_o}. \tag{79}$$

We complete the proof by contradiction. Assume, the desired result does not hold for the first $T$ iterations, i.e.,

$$\mathbb{E}[\|\nabla V_\zeta(\theta_k)\|^2] \geq \frac{2G_V^2 L_V \beta}{BD_o} + \epsilon^2 \tag{80}$$

for any $0 \leq k \leq T - 1$. Then, by (79), for any $0 \leq k \leq T - 1$, we have

$$-\mathbb{E}[V_\zeta(\theta_{k+1})] \leq -\mathbb{E}[V_\zeta(\theta_k)] - \frac{\beta\epsilon^2}{2} - \frac{G_V^2 L_V \beta^2}{2BD_o}. \tag{81}$$

Adding up this result for $k = 0, ..., T - 1$ yields

$$-\mathbb{E}[V_\zeta(\theta_T)] \leq -\mathbb{E}[V_\zeta(\theta_0)] - T\left(\frac{\beta\epsilon^2}{2} + \frac{G_V^2 L_V \beta^2}{2BD_o}\right). \tag{82}$$

Note that, by Assumption 1, both $\mathbb{E}[V_\zeta(\theta_T)]$ and $\mathbb{E}[V_\zeta(\theta_0)]$ have values between zero and $R$, and thus, their difference is bounded by $R$. Therefore,

$$T\left(\frac{\beta\epsilon^2}{2} + \frac{G_V^2 L_V \beta^2}{2BD_o}\right) \leq R \tag{83}$$

which gives us the desired result. $\qquad\square$

# H  More Details on the Numerical Experiment Section

In this section of the Appendix we detail our experimental setup beyond the description given in Section 5. We use a neural network policy with two 100-unit hidden layers and ReLU activations. For simplicity, we use vanilla policy gradient (VPG) for both the inner adaption steps and the outer meta steps.

In all cases, we train both algorithms for 500 (meta-)epochs, using a meta-batch size of 20 tasks for 2D-navigation and 40 tasks for the locomotion one. For all tasks, we use 20 episodes per adaptation step. All rewards are discounted with a factor $\gamma = 0.99$. We use a horizon $H = 100$ for 2D-navigation and $H = 200$ for locomotion tasks. Next, we use a learning rate of 0.1 for the inner steps, and 0.001 for the outer ones. Finally, all experiments are averaged over 10 random seeds.

The MuJoCo locomotion environments we consider are

- **Half-Cheetah Random Direction** which simulates the dynamics of a "cheetah" robot which is trained to move fast. In this environment, each task is a goal direction (forward/backward) and the reward at each timestep is given by the magnitude of the agent's velocity.

- **Half-Cheetah Random Velocity** which uses the same "cheetah" robot, but now each task is a goal velocity. The reward at each timestep is given by the negative of the absolute difference between the current and goal velocities.

- **Swimmer Random Velocity** which simulates the dynamics of a planar "swimmer" robot in a viscous liquid. The swimmer needs to use viscous drag to propel itself. Like with the other direction environment, each task is a goal direction (forward/backward) and the reward at each timestep is given by the magnitude of the agent's velocity.

For each of the environments, we present results using 1, 2, and 3 gradient steps.

Finally, we use MuJoCo [26] license and perform all experiments on an internal server using 2 NVIDIA V100 GPUs.