# OpenReview forum: "On the Convergence Theory of Debiased Model-Agnostic Meta-Reinforcement Learning"
_NeurIPS.cc/2021/Conference — NeurIPS 2021 Poster_

### Official Review · Reviewer_43di · 2021-07-14

**Rating:** 7
**Confidence:** 5

**Summary:**

This paper proposed a new debiased model-agnostic meta-reinforcement learning algorithm. The main improvement is that the paper considers a more accurate setting where one does not have accurate estimate on the gradient for each task, which is assumed by previous works. One has to estimate the gradient from samples in each task, and the samples themselves depend on the global parameter $\theta$, there will be an extra term for the global gradient.  The authors hence write the exact gradient and show the convergence. Numerical results show good performance.

**Limitations And Societal Impact:**

Despite the improvement shown in the numerical experiments, it would be sound if the authors can theoretically show the advantage of the proposed method versus the methods in the literature. To be more specific, how much does assuming the knowledge of the second-order information of the expected reward hurt the actual performance, when the information is not available. Will it cause bias or increased variance?

**Main Review:**

The paper makes a solid improvement over the previous methods by considering a more accurate setting, i.e. the gradient of each task is estimated from batch data. By considering the more accurate setting, they are also able to achieve a better numerical performance.

The paper is well-written and easy to follow.

**Time Spent Reviewing:**

1.5

---

> ### Author Response · Authors · 2021-08-11
> **Response to Reviewer 43di**
>
> We thank the reviewer for the constructive feedback. In the following paragraphs we briefly address the issues raised by the reviewer.
>
> **Comment:** Despite the improvement shown in the numerical experiments, it would be sound if the authors can theoretically show the advantage of the proposed method versus the methods in the literature. To be more specific, how much does assuming the knowledge of the second-order information of the expected reward hurt the actual performance, when the information is not available. Will it cause bias or increased variance?
>
>
> **Response:** This is a good point. Note that our proposed SG-MRL method for solving the loss in (6) requires access to second-order information of the expected reward. Dropping the second-order term in the algorithm would lead to a bias in the descent direction and could lower the quality of obtained solution with the same choice of stepsize. In fact, in [14], the authors study this phenomenon for the supervised learning setting and study the effect of bias induced by ignoring the second-order term. Extending that analysis to the Meta-RL setting is indeed an interesting problem that we plan to investigate, but it is beyond the scope of the current submission. We will discuss the impact of dropping the second-order term in the revised paper and will include a pointer to the work in [14] for the supervised learning setting. Thanks for raising this point.

---

### Official Review · Reviewer_r6Av · 2021-07-14

**Rating:** 6
**Confidence:** 3

**Summary:**

This paper aims to tackle the non-diminishing bias in MAMRL (MAML for RL) due to the stochastic policy gradient update for subtasks. Such bias hinders the first-order optimality when solving MAML with gradient descent. The authors compute the exact gradient of the MAMRL objective and identify the additional term that arises in RL. The authors then estimate such a gradient with an unbiased approximation based on samples and propose SG-MRL that utilizes such approximation to solve MAMRL. The authors show that SG-MRL converges to first-order stationary points under the standard assumptions and further conduct experiments on SG-MRL against baselines in the MuJoCo environment.

**Limitations And Societal Impact:**

See inquiries in the Main review. I do not have concerns regarding the societal impact of this work.


**Main Review:**

This paper is a fairly solid work on MAML, with both theoretical insights and empirical improvements over baselines. The algorithm is simple and well-motivated, and the theory is reasonable and well-explained. Experiments demonstrate the improvement of the proposed SG-MRL against baselines, which also partially corroborates the theory.

It would be better if the authors could provide a more comprehensive comparison of SG-MRL against MAML on more tasks. Still, this is a minor issue. The main point of this work is on the theoretical side, for which the experiments are sufficient to me.

Some inquiries:

1. Is the bias correction term only adaptable to the update proposed in (6)? As the authors pointed out, the MAML gradient in (16) is biased and would also require a similar term to the bias correction term of SG-MRL. Does the difference in the objective (between (6) and (14)) affect the formulation of the bias correction term?

2. In lines 235-244, the authors mentioned that having a biased gradient requires careful selection of stepsize for MAML, while SG-MRL does not suffer from such stepsize selection. Could the authors provide more details regarding this point?

3. The theory involves the inner learning rate $\alpha$ as a smoothness parameter of the objective and requires $\alpha$ to be small for convergence. Can the theory say something about the effect of $\alpha$ on convergence and the overall performance of SG-MRL?

**Time Spent Reviewing:**

2

---

> ### Author Response · Authors · 2021-08-11
> **Response to Reviewer r6Av**
>
> We thank the reviewer for the constructive feedback. In the following paragraphs we briefly address the issues raised by the reviewer.
>
> **Comment:**
> Is the bias correction term only adaptable to the update proposed in (6)? As the authors pointed out, the MAML gradient in (16) is biased and would also require a similar term to the bias correction term of SG-MRL. Does the difference in the objective (between (6) and (14)) affect the formulation of the bias correction term?
>
>
> **Response:** This is an excellent point. Note that, in our paper, we propose a new formulation (Eq. (6)) which we believe is more realistic for RL applications as it assumes using stochastic gradient for updating the model at test time, in contrast to MAML formulation in (14) which assumes using exact gradients for updating the model. For this formulation, i.e, Eq. (6), our new algorithm (SG-MRL) finds an unbiased estimator of gradient's loss.
>
> However, let us assume we are interested in solving the original MAML formulation, i.e., Eq. (14), even though it is less practical. Even in this case, as noted by the reviewer and mentioned in the paper, MAML gradient update (16) is a biased estimator for the gradient of MAML objective function (14). Controlling this bias has been studied in the context of supervised learning, see, e.g. [Hu et al., 2020]. However, this is out of the scope of this work as we mainly focus on the other formulation (6).
>
> $~$
>
> **Comment:**
> In lines 235-244, the authors mentioned that having a biased gradient requires careful selection of stepsize for MAML, while SG-MRL does not suffer from such stepsize selection. Could the authors provide more details regarding this point?
>
> **Response:**
> Note that, in general optimization analyses, when we deal with unbiased stochastic gradient estimators (similar to the case of SG-MRL in RL setting), we could converge to the exact solution by using a diminishing or small stepsize. However, when we have access to biased gradient estimators (as in MAML setting), even with diminishing or small stepsize we might only converge to a neighborhood of the optimal solution, where the radius of our convergence depends on the bias. To resolve this issue, one needs to control the bias in the gradient directions and lower the bias as time progresses using some debiasing techniques. For instance, the work in [Hu et al., 2020] studies this problem in detail for debiasing MAML in the supervised learning setting. We will highlight this point in the revised paper. Thank you for raising this point.
>
> $~$
>
> **Comment:**
> The theory involves the inner learning rate $\alpha$ as a smoothness parameter of the objective and requires $\alpha$ to be small for convergence. Can the theory say something about the effect of $\alpha$ on convergence and the overall performance of SG-MRL?
>
>
> **Response:**
> Thanks for the great question. First, recall that $\zeta$ is the number of gradient ascent steps that we take at test time to update the model. As stated in our paper and Theorem 3 in the supplementary material, the smoothness parameter grows exponentially with $\zeta$. However, The exponential term $2^\zeta$ in smoothness parameter in equation (49) is just simplification of $(1+\eta_H \alpha)^\zeta$, as explained in lines 604-606. That means that having larger $\alpha$ could lead to a larger gradient Lipschitz parameter of the MAML objective function. This would mean that we need to choose a smaller learning rate to train the MAML model, and hence, the algorithm will convergence slower. However, on the other side, larger $\alpha$ means that we take a larger update at test time and this *could potentially* lead to a better performance for the updated model. However, studying this trade-off is out of the scope of this paper, and in fact, to the best of our knowledge, it has not been studied even in the context of supervised learning.
>
>
> [Hu et al., 2020] Hu, Y., Zhang, S., Chen, X., and He, N. (2020).  Biasedstochastic first-order methods for conditional stochastic optimization andapplications in meta learning.Advances in Neural Information ProcessingSystems, 33.

---

### Official Review · Reviewer_yqwx · 2021-07-16

**Rating:** 7
**Confidence:** 3

**Summary:**

This paper studies the Model-Agnostic Meta-Reinforcement-Learning (MAMRL) from a theoretical perspective. In particular, it proposes a modified MAMRL objective along with its unbiased gradient estimator, establishing a new Meta-RL algorithm called SR-MRL. It theoreticallly proves the convergence of SR-MRL and experimentally shows its comparable performance to the original MAML algorithm.

**Limitations And Societal Impact:**

The authors adequately addressed the limitations of their work in Section 6 by comments on their assumptions. The potential negative social impact is not addressed, which will generally be all potential malicious usage of any reinforcement learning algorithms.

**Main Review:**

Overall, alghouth the techniques used in this paper do not seem to be very novel, I consider its results as a significant progress of Meta-RL because it is the first provably convergent Meta-RL algorithm, which meanwhile also achieves comparable performance to the MAML algorithm. The whole paper is also very well-written with clear logic flow.

One concern I have is about the assumptions this paper use. It seems that the convergence does not rely on any assumptions about underlying MDPs at all, which looks a little bit wired to me. Did I miss anything?

Questions:
- Is the exponential dependency of $V_{\zeta}$'s smoothness parameter in $\zeta$ fundamental? Is there any intuitive explanation on why its dependency is exponential?
- For the Remark 2, is there any benefits of increasing $\zeta$ from a theoretical perspective?
- Does the convergence result rely on any assumptions about the transition kernel of $\mathcal{M}_i$ or it can be completely arbitrary?

Minor suggestions on writing:
- At line 301 on page 8, *Theorem 1* -> *Proposition 1*

---

Score increased after rebuttal.

**Time Spent Reviewing:**

6

---

> ### Author Response · Authors · 2021-08-11
> **Response to Reviewer yqwx**
>
> We thank the reviewer for the constructive feedback. In the following paragraphs we briefly address the issues raised by the reviewer.
>
> **Comment:** One concern I have is about the assumptions this paper use. It seems that the convergence does not rely on any assumptions about underlying MDPs at all, which looks a little bit wired to me. Did I miss anything? Does the convergence result rely on any assumptions about the transition kernel of $\mathcal{M}_i$ or it can be completely arbitrary?
>
>
> **Response:**
> It is worth noting that we in fact have an assumption on the underlying MDP as Assumption 1 is on the reward function which is a part of MDP definition. However, the reviewer is right that we do not have any assumption on the underlying transition Kernel or initial distribution. In fact, as shown in [22] and [23], Assumptions 1 and 2 are enough to provide convergence to first order stationary points for gradient policy methods in RL theory and we do not need further assumptions there as well.
>
> $~$
>
> **Comment:** Is the exponential dependency of $V_\zeta$'s smoothness parameter in $\zeta$ fundamental? Is there any intuitive explanation on why its dependency is exponential?
>
>
> **Response:** Note that the smoothness parameter of the function $V_\zeta$ for the case of $\zeta>1$ can be written as $(1+\eta_H \alpha)^\zeta$ and since we select our stepsize such that $\alpha \eta_H<1$, we can show that the smoothness parameter is bounded above by $2^\zeta$. We'll highlight this point in the revised paper.
>
> We should also add that the exponential growth is  inevitable without further assumptions. To better highlight this matter, let us consider a simple maximization problem.  Assume that we have only one task with reward function $f(x) = (1/2) x^2$ and our goal is to maximize this function over a compact set. In this case, the corresponding MAML objective function $F_\zeta(x) = f(\Psi(\Psi(... \Psi(x))))$ where $\Psi(x) = x + \alpha \nabla f(x)$. In this case, it is straightforward to see that $F_\zeta(x) = (1/2) (1+ \alpha)^{2\zeta} x^2$, and hence the smoothness parameter will be equal to $ (1+ \alpha)^{2\zeta}$.
>
> $~$
>
> **Comment:** For the Remark 2, is there any benefits of increasing $\zeta$ from a theoretical perspective?
>
> **Response:** This is a good point. Note that, as discussed in Remark 2, there is an inherent trade-off in increasing $\zeta$: Larger $\zeta$ means solving the maximization problem would be harder as the smoothness parameter could be larger, however, on the positive side, we take more steps to update the policy at test time, and hence this could lead to a better performance over the test task. Studying this trade-off theoretically is a very interesting yet open question in meta-learning literature (and even for the case of supervised learning), and we see it as a potential future direction.

---

> > ### Comment · Reviewer_yqwx · 2021-08-20
> > **Reply to Authors' Response**
> >
> > Thank you very much for your detailed response! My concerns have been well-addressed and I'll also increase my score.

---

### Official Review · Reviewer_WxkP · 2021-07-16

**Rating:** 7
**Confidence:** 4

**Summary:**

This paper deals with the problem of few-shot meta learning, i.e. where one explicitly optimizes for the performance of doing one (or more) steps of SGD on a new task. This was intially introduced through the MAML method. However, the authors argue that MAML only discusses gradient descent while in practise, approximations lead to doing stochastic gradient desecent instead, thus making the original formulation of the problem biased. To that end, this paper proposes an SGD based meta learning objective, which ensures simpler convergence analysis as well as better performance in two benchmark problems from the RL setting.

**Ethical Concerns:**

I do not think there any explicit ethical concerns raised by this work.

**Limitations And Societal Impact:**

Suggestions: My sole primary suggestion would be to test this method in the supervised learning setting as well, just as in the MAML paper, so as to really drive the point home that correcting for the bias can have considerable empirical gains. This will again highlight that the paper does not only contribute theoretically.

I do not think there any explicit negative societal impacts of this work.

**Main Review:**

Originality: The motivation of the work is solid in my opinion. The main question the authors ask is indeed important and the authors do a good job at highlighting it.

Quality: The overall quality of the work is quite high. The explanations of the SG-MRL theory are well detailed and intuitive.

Clarity: The paper reads quite clearly overall. The flow of content is pretty nice as well.

Significance: I believe the paper contributes in both fronts, theoretically and empirically. In the theory part, it sheds important insight on the issues with prior work and how those can be fixed, in the case when we do not have access to exact gradients. In the practical part, the results are quite strong and show that a simple fix can provide better gains. It would be even better in my opinion if there were some ablations on how much the gradient is biased by (across training) and how/if it matches with the reward curves.

Questions/Clarifications:

- Going through the MAML paper, it seems like the authors proposed an objective that can be optimized by auto-diff end-to-end. Therefore, they did not use the inner/outer loops explicitly to explain the application of gradients in their work. Is it accurate to say that SG-MRL can also be used with auto-diff directly and that the inner/outer loop computations happen the same way as described in the paper for both SG-MRL and MAML. Or is it that because MAML induces a biased gradient, it can work with auto-diff but SG-MRL cannot, since it requires computing independent averages at both the inner and outer loops?

- The empirical results look really convincing, in that computing an unbiased estimate of the stochastic gradient can have considerable difference in performance. However, the overall story of the paper leans heavily towards the theoretical contributions. Is there any specific reason why? Is the implementation more complicated than MAML in any sense?

**Time Spent Reviewing:**

5 hrs

---

> ### Author Response · Authors · 2021-08-11
> **Response to Reviewer WxkP**
>
> We thank the reviewer for the constructive feedback. In the following paragraphs we briefly address the issues raised by the reviewer.
>
> **Comment:** Going through the MAML paper, it seems like the authors proposed an objective that can be optimized by auto-diff end-to-end. Therefore, they did not use the inner/outer loops explicitly to explain the application of gradients in their work. Is it accurate to say that SG-MRL can also be used with auto-diff directly and that the inner/outer loop computations happen the same way as described in the paper for both SG-MRL and MAML. Or is it that because MAML induces a biased gradient, it can work with auto-diff but SG-MRL cannot, since it requires computing independent averages at both the inner and outer loops?
>
>
> **Response:**
> The reviewer is indeed correct that both MAML and SG-MRL can be optimized directly with automatic differentiation, and thus can be implemented in any auto-diff library (e.g. PyTorch, TensorFlow, etc.). The inner/outer loop computations in SG-MRL happen in an identical way to MAML and thus switching over to SG-MRL is straightforward from an implementation perspective. We will highlight this point in the revised paper.
>
> $~$
>
> **Comment:**
> The empirical results look really convincing, in that computing an unbiased estimate of the stochastic gradient can have considerable difference in performance. However, the overall story of the paper leans heavily towards the theoretical contributions. Is there any specific reason why? Is the implementation more complicated than MAML in any sense?
>
>
> **Response:**
> As the reviewer has correctly pointed out, the main focus of this paper is on developing and studying a meta-RL algorithm that is *provably* convergent and does not suffer from the issue of bias directions (as in MAML). In addition to our theoretical study, we numerically investigated the performance of SG-MRL and highlight the issue of bias directions in MAML and how SG-MRL is able to improve the MAML performance. Also, we would like to highlight that SG-MRL is straightforward to implement, and we have provided the implementation in the supplementary file. In addition, as stated in our paper, in our experiments SG-MRL is at most $1.05$ times slower than MAML (due to the additional term in its update), and hence there is no specific obstacle in running SG-MRL in other RL environments.
>
> $~$
>
> **Comment:**
> Suggestions: My sole primary suggestion would be to test this method in the supervised learning setting as well, just as in the MAML paper, so as to really drive the point home that correcting for the bias can have considerable empirical gains. This will again highlight that the paper does not only contribute theoretically.
>
> **Response:**
> We want to highlight that the additional term in SG-MRL update compared to MAML (see Algorithm 3) comes from the fact that the distribution of trajectories depends on the policy parameter, and hence, the derivative of the objective function generates an additional term (see line 223 for further explanation). However, in supervised learning, typically the underlying distribution is independent of the model, and hence, we do not have such additional term.

---

> > ### Comment · Reviewer_WxkP · 2021-08-19
> > **Re: Response**
> >
> > Thanks for answering my questions. I am pretty satisfied with the answers and do not have any major concerns. Thus, I'm keeping to my original score. I hope the authors release the code soon so other meta-RL methods can build over it. Maybe highlighting a few-line change in the original MAML code for RL would also be useful in faster adoption of SG-MRL.

---

### Public Comment · ~Jin_Zhang6 · 2022-06-13
**Any connections between SG-MRL and ProMP?**

Hi there, thank you for your interesting work!

I've got a question. What's the connection between the additional term in SG-MRL and $\nabla_\theta J_{pre}(\tau,\tau')$ in Eq. (1) in ProMP (https://arxiv.org/pdf/1810.06784.pdf?ref=https://githubhelp.com)? Are they addressing the same problem? Is the additional term in SG-MRL dealing with $\theta_k$'s influence on $D_{in}^{i}$? If not, what is its meaning?

Looking forward to your reply. Thank you very much!

---

### Decision · Program_Chairs · 2021-09-27

**Decision:**

Accept (Poster)

**Comment:**

All reviews find that this paper is a solid contribution making a reasonable contribution to the empirical as well as the theoretical state-of-the-art of MAML, so that I recommend to accept it. Please take into account the points raised in the reviews when preparing the final version.